# A*Net: A Scalable Path-based Reasoning Approach for Knowledge Graphs

**Zhaocheng Zhu[1,2,∗], Xinyu Yuan[1,2,∗], Mikhail Galkin[3,†], Sophie Xhonneux[1,2]**
**Ming Zhang[4], Maxime Gazeau[5], Jian Tang[1,6,7]**
[1]Mila - Québec AI Institute, [2]University of Montréal
[3]Intel AI Lab, [4]Peking University, [5]LG Electronics AI Lab
[6]HEC Montréal, [7]CIFAR AI Chair

## Abstract

Reasoning on large-scale knowledge graphs has been long dominated by embedding methods. While path-based methods possess the inductive capacity that embeddings lack, their scalability is limited by the exponential number of paths. Here we present A*Net, a scalable path-based method for knowledge graph reasoning. Inspired by the A* algorithm for shortest path problems, our A*Net learns a priority function to select important nodes and edges at each iteration, to *reduce time and memory footprint for both training and inference*. The ratio of selected nodes and edges can be specified to trade off between performance and efficiency. Experiments on both transductive and inductive knowledge graph reasoning benchmarks show that A*Net achieves competitive performance with existing state-of-the-art path-based methods, while merely visiting 10% nodes and 10% edges at each iteration. On a million-scale dataset ogbl-wikikg2, A*Net not only achieves a new state-of-the-art result, but also converges faster than embedding methods. A*Net is the first path-based method for knowledge graph reasoning at such scale.

## 1 Introduction

Reasoning, the ability to apply logic to draw new conclusions from existing facts, has been long pursued as a goal of artificial intelligence [32, 20]. Knowledge graphs encapsulate facts in relational edges between entities, and serve as a foundation for reasoning. Reasoning over knowledge graphs is usually studied in the form of knowledge graph completion, where a model is asked to predict missing triplets based on observed triplets in the knowledge graph. Such a task can be used to not only populate existing knowledge graphs, but also improve downstream applications like multi-hop logical reasoning [34], question answering [5] and recommender systems [53].

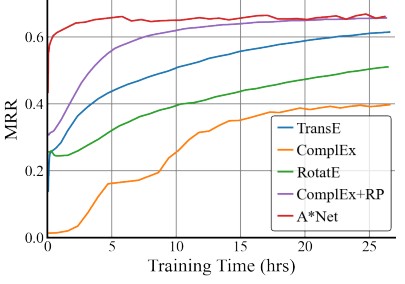

Figure 1: Validation MRR w.r.t. training time on ogbl-wikikg2 (1 A100 GPU). A*Net achieves state-of-the-art performance and the fastest convergence.

One challenge central to knowledge graph reasoning is the scalability of reasoning methods, as many real-world knowledge graphs [2, 44] contain millions of entities and triplets. Typically, large-scale knowledge graph reasoning is solved by embedding methods [6, 42, 38], which learn an embedding for each entity and relation to reconstruct the structure of the knowledge

---

[∗]Equal contribution. Code is available at `https://github.com/DeepGraphLearning/AStarNet`
[†]Work done while at Mila - Québec AI Institute.

37th Conference on Neural Information Processing Systems (NeurIPS 2023).

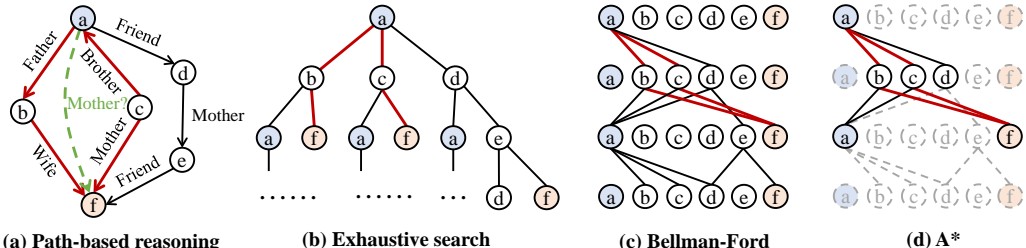

(a) **Path-based reasoning**     (b) **Exhaustive search**     (c) **Bellman-Ford**     (d) **A\***

Figure 2: **(a)** Given a query $(a, Mother, ?)$, only a few important paths (showed in red) are necessary for reasoning. Note that paths can go in the reverse direction of relations. **(b)** Exhaustive search algorithm (e.g., Path-RNN, PathCon) enumerates all paths in exponential time. **(c)** Bellman-Ford algorithm (e.g., NeuralLP, DRUM, NBFNet, RED-GNN) computes all paths in polynomial time, but needs to propagate through all nodes and edges. **(d)** A\*Net learns a priority function to select a subset of nodes and edges at each iteration, and avoids exploring all nodes and edges.

graph. Due to its simplicity, embedding methods have become the *de facto* standard for knowledge graphs with millions of entities and triplets. With the help of multi-GPU embedding systems [57, 56], they can further scale to knowledge graphs with billions of triplets.

Another stream of works, path-based methods [28, 31, 11, 58], predicts the relation between a pair of entities based on the paths between them. Take the knowledge graph in Fig. 2(a) as an example, we can prove that *Mother(a, f)* holds, because there are two paths $a \xrightarrow{Father} b \xrightarrow{Wife} f$ and $a \xleftarrow{Brother} c \xrightarrow{Mother} f$. As the semantics of paths are purely determined by relations rather than entities, path-based methods naturally generalize to unseen entities (i.e., inductive setting), which cannot be handled by embedding methods. However, the number of paths grows exponentially w.r.t. the path length, which hinders the application of path-based methods on large-scale knowledge graphs.

Here we propose A\*Net to tackle the scalability issue of path-based methods. The key idea of our method is to search for important paths rather than use all possible paths for reasoning, thereby reducing time and memory in training and inference. Inspired by the A\* algorithm [22] for shortest path problems, given a head entity $u$ and a query relation $q$, we compute a priority score for each entity to guide the search towards more important paths. At each iteration, we select $K$ nodes and $L$ edges according to their priority, and use message passing to update nodes in their neighborhood. Due to the complex semantics of knowledge graphs, it is hard to use a handcrafted priority function like the A\* algorithm without a significant performance drop (Tab. 6a). Instead, we design a neural priority function based on the node representations at the current iteration, which can be end-to-end trained by the objective function of the reasoning task without any additional supervision.

We verify our method on 4 transductive and 2 inductive knowledge graph reasoning datasets. Experiments show that A\*Net achieves competitive performance against state-of-the-art path-based methods on FB15k-237, WN18RR and YAGO3-10, even with only 10% of nodes and 10% edges at each iteration (Sec. 4.2). To verify the scalability of our method, we also evaluate A\*Net on ogbl-wikikg2, a million-scale knowledge graph that is 2 magnitudes larger than datasets solved by previous path-based methods. Surprisingly, with only 0.2% nodes and 0.2% edges, our method outperforms existing embedding methods and establishes new state-of-the-art results (Sec. 4.2) as the first non-embedding method on ogbl-wikikg2. By adjusting the ratios of selected nodes and edges, one can trade off between performance and efficiency (Sec. 4.3). A\*Net also converges significantly faster than embedding methods (Fig. 1), which makes it a promising model for deployment on large-scale knowledge graphs. Additionally, A\*Net offers interpretability that embeddings do not possess. Visualization shows that A\*Net captures important paths for reasoning (Sec. 4.4).

## 2   Preliminary

**Knowledge Graph Reasoning**   A knowledge graph $\mathcal{G} = (\mathcal{V}, \mathcal{E}, \mathcal{R})$ consists of sets of entities (nodes) $\mathcal{V}$, facts (edges) $\mathcal{E}$ and relation types $\mathcal{R}$. Each fact is a triplet $(x, r, y) \in \mathcal{V} \times \mathcal{R} \times \mathcal{V}$, which indicates a relation $r$ from entity $x$ to entity $y$. The task of knowledge graph reasoning aims at answering queries like $(u, q, ?)$ or $(?, q, u)$. Without loss of generality, we assume the query is

$(u, q, ?)$, since $(?, q, u)$ equals to $(u, q^{-1}, ?)$ with $q^{-1}$ being the inverse of $q$. Given a query $(u, q, ?)$, we need to predict the answer set $\mathcal{V}_{(u,q,?)}$, such that $\forall v \in \mathcal{V}_{(u,q,?)}$ the triplet $(u, q, v)$ should be true.

**Path-based Methods**   Path-based methods [28, 31, 11, 58] solve knowledge graph reasoning by looking at the paths between a pair of entities in a knowledge graph. For example, a path $a \xrightarrow{Father} b \xrightarrow{Wife} f$ may be used to predict *Mother(a, f)* in Fig. 2(a). From a representation learning perspective, path-based methods aim to learn a representation $\boldsymbol{h}_q(u, v)$ to predict the triplet $(u, q, v)$ based on all paths $\mathcal{P}_{u \rightsquigarrow v}$ from entity $u$ to entity $v$. Following the notation in [58][3], $\boldsymbol{h}_q(u, v)$ is defined as

$$\boldsymbol{h}_q(u, v) = \bigoplus_{P \in \mathcal{P}_{u \rightsquigarrow v}} \boldsymbol{h}_q(P) = \bigoplus_{P \in \mathcal{P}_{u \rightsquigarrow v}} \bigotimes_{(x,r,y) \in P} \boldsymbol{w}_q(x, r, y) \tag{1}$$

where $\bigoplus$ is a permutation-invariant aggregation function over paths (e.g., sum or max), $\bigotimes$ is an aggregation function over edges that may be permutation-sensitive (e.g., matrix multiplication) and $\boldsymbol{w}_q(x, r, y)$ is the representation of triplet $(x, r, y)$ conditioned on the query relation $q$. $\bigotimes$ is computed before $\bigoplus$. Typically, $\boldsymbol{w}_q(x, r, y)$ is designed to be independent of the entities $x$ and $y$, which enables path-based methods to generalize to the inductive setting. However, it is intractable to compute Eqn. 1, since the number of paths usually grows exponentially w.r.t. the path length.

**Path-based Reasoning with Bellman-Ford algorithm**   To reduce the time complexity of path-based methods, recent works [50, 35, 58, 54] borrow the Bellman-Ford algorithm [4] from shortest path problems to solve path-based methods. Instead of enumerating each possible path, the Bellman-Ford algorithm iteratively propagates the representations of $t - 1$ hops to compute the representations of $t$ hops, which achieves a polynomial time complexity. Formally, let $\boldsymbol{h}_q^{(t)}(u, v)$ be the representation of $t$ hops. The Bellman-Ford algorithm can be written as

$$\boldsymbol{h}_q^{(0)}(u, v) \leftarrow \mathbb{1}_q(u = v) \tag{2}$$

$$\boldsymbol{h}_q^{(t)}(u, v) \leftarrow \boldsymbol{h}_q^{(0)}(u, v) \oplus \bigoplus_{(x,r,v) \in \mathcal{E}(v)} \boldsymbol{h}_q^{(t-1)}(u, x) \otimes \boldsymbol{w}_q(x, r, v) \tag{3}$$

where $\mathbb{1}_q$ is a learnable indicator function that defines the representations of 0 hops $\boldsymbol{h}_q^{(0)}(u, v)$, also known as the boundary condition of the Bellman-Ford algorithm. $\mathcal{E}(v)$ is the neighborhood of node $v$. Despite the polynomial time complexity achieved by the Bellman-Ford algorithm, Eqn. 3 still needs to visit $|\mathcal{V}|$ nodes and $|\mathcal{E}|$ edges to compute $\boldsymbol{h}_q^{(t)}(u, v)$ for all $v \in \mathcal{V}$ in each iteration, which is not feasible for large-scale knowledge graphs.

**A\* Algorithm**   A\* algorithm [22] is an extension of the Bellman-Ford algorithm for shortest path problems. Unlike the Bellman-Ford algorithm that propagates through every node uniformly, the A\* algorithm prioritizes propagation through nodes with higher priority according to a heuristic function specified by the user. With an appropriate heuristic function, A\* algorithm can reduce the search space of paths. Formally, with the notation from Eqn. 1, the priority function for node $x$ is

$$s(x) = d(u, x) \otimes g(x, v) \tag{4}$$

where $d(u, x)$ is the length of current shortest path from $u$ to $x$, and $g(x, v)$ is a heuristic function estimating the cost from $x$ to the target node $v$. For instance, for a grid-world shortest path problem (Fig. 4(a)), $g(x, v)$ is usually defined as the $L_1$ distance from $x$ to $v$, $\otimes$ is the addition operator, and $s(x)$ is a lower bound for the shortest path length from $u$ to $v$ through $x$. During each iteration, the A\* algorithm prioritizes propagation through nodes with smaller $s(x)$.

## 3   Proposed Method

We propose A\*Net to scale up path-based methods with the A\* algorithm. We show that the A\* algorithm can be derived from the observation that only a small set of paths are important for reasoning (Sec. 3.1). Since it is hard to handcraft a good priority function for knowledge graph reasoning (Tab. 6a), we design a neural priority function, and train it end-to-end for reasoning (Sec. 3.2).

---

[3]$\oplus$ and $\otimes$ are binary operations (akin to $+$, $\times$), while $\bigoplus$ and $\bigotimes$ are n-ary operations (akin to $\sum$, $\prod$).

### 3.1 Path-based Reasoning with A* Algorithm

As discussed in Sec. 2, the Bellman-Ford algorithm visits all $|\mathcal{V}|$ nodes and $|\mathcal{E}|$ edges. However, in real-world knowledge graphs, only a small portion of paths is related to the query. Based on this observation, we introduce the concept of important paths. We then show that the representations of important paths can be iteratively computed with the A* algorithm under mild assumptions.

**Important Paths for Reasoning** Given a query relation and a pair of entities, only some of the paths between the entities are important for answering the query. Consider the example in Fig. 2(a), the path $a \xrightarrow{\textit{Friend}} d \xrightarrow{\textit{Mother}} e \xrightarrow{\textit{Friend}} f$ cannot determine whether $f$ is an answer to *Mother(a, ?)* due to the use of the *Friend* relation in the path. On the other hand, kinship paths like $a \xrightarrow{\textit{Father}} b \xrightarrow{\textit{Wife}} f$ or $a \xleftarrow{\textit{Brother}} c \xrightarrow{\textit{Mother}} f$ are able to predict that *Mother(a, f)* is true. Formally, we define $\mathcal{P}_{u \rightsquigarrow v|q} \subseteq \mathcal{P}_{u \rightsquigarrow v}$ to be the set of paths from $u$ to $v$ that is important to the query relation $q$. Mathematically, we have

$$\boldsymbol{h}_q(u,v) = \bigoplus_{P \in \mathcal{P}_{u \rightsquigarrow v}} \boldsymbol{h}_q(P) \approx \bigoplus_{P \in \mathcal{P}_{u \rightsquigarrow v|q}} \boldsymbol{h}_q(P) \tag{5}$$

In other words, any path $P \in \mathcal{P}_{u \rightsquigarrow v} \setminus \mathcal{P}_{u \rightsquigarrow v|q}$ has negligible contribution to $\boldsymbol{h}_q(u,v)$. In real-world knowledge graphs, the number of important paths $|\mathcal{P}_{u \rightsquigarrow v|q}|$ may be several orders of magnitudes smaller than the number of paths $|\mathcal{P}_{u \rightsquigarrow v}|$ [11]. If we compute the representation $\boldsymbol{h}_q(u,v)$ using only the important paths, we can scale up path-based reasoning to large-scale knowledge graphs.

**Iterative Computation of Important Paths** Given a query $(u, q, ?)$, we need to discover the set of important paths $\mathcal{P}_{u \rightsquigarrow v|q}$ for all $v \in \mathcal{V}$. However, it is challenging to extract important paths from $\mathcal{P}_{u \rightsquigarrow v}$, since the size of $\mathcal{P}_{u \rightsquigarrow v}$ is exponentially large. Our solution is to explore the structure of important paths and compute them iteratively. We first show that we can cover important paths with iterative path selection (Eqn. 6 and 7). Then we approximate iterative path selection with iterative node selection (Eqn. 8).

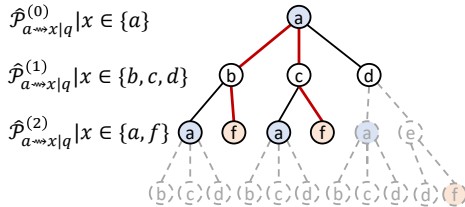

Figure 3: The colored paths are important paths $\mathcal{P}_{u \rightsquigarrow v|q}$, while the solid paths are the superset $\hat{\mathcal{P}}_{u \rightsquigarrow v|q}$ used in Eqn. 7.

Notice that paths in $\mathcal{P}_{u \rightsquigarrow v}$ form a tree structure (Fig. 3). On the tree, a path is not important if any prefix of this path is not important for the query. For example, in Fig. 2(a), $a \xrightarrow{\textit{Friend}} d \xrightarrow{\textit{Mother}} e \xrightarrow{\textit{Friend}} f$ is not important, as its prefix $a \xrightarrow{\textit{Friend}} d$ is not important for the query *Mother*. Therefore, we assume there exists a path selection function $m_q : 2^{\mathcal{P}} \mapsto 2^{\mathcal{P}}$ that selects important paths from a set of paths given the query relation $q$. $2^{\mathcal{P}}$ is the set of all subsets of $\mathcal{P}$. With $m_q$, we construct the following set of paths $\hat{\mathcal{P}}^{(t)}_{u \rightsquigarrow v|q}$ iteratively

$$\hat{\mathcal{P}}^{(0)}_{u \rightsquigarrow v|q} \leftarrow \{(u, \text{self loop}, v)\} \text{ if } u = v \text{ else } \varnothing \tag{6}$$

$$\hat{\mathcal{P}}^{(t)}_{u \rightsquigarrow v|q} \leftarrow \bigcup_{\substack{x \in \mathcal{V} \\ (x,r,v) \in \mathcal{E}(v)}} \left\{ P + \{(x,r,v)\} \Big| P \in m_q(\hat{\mathcal{P}}^{(t-1)}_{u \rightsquigarrow x|q}) \right\} \tag{7}$$

where $P + \{(x,r,v)\}$ concatenates the path $P$ and the edge $(x,r,v)$. The paths $\hat{\mathcal{P}}^{(t)}_{u \rightsquigarrow v|q}$ computed by the above iteration is a superset of the important paths $\mathcal{P}^{(t)}_{u \rightsquigarrow v|q}$ of length $t$ (see Thm. A.1 in App. A). Due to the tree structure of paths, the above iterative path selection still requires exponential time. Hence we further approximate iterative path selection with iterative node selection, by assuming paths with the same length and the same stop node can be merged. The iterative node selection replacing Eqn. 7 is (see Prop. A.3 in App. A)

$$\hat{\mathcal{P}}^{(t)}_{u \rightsquigarrow v|q} \leftarrow \bigcup_{\substack{x \in n^{(t-1)}_{uq}(\mathcal{V}) \\ (x,r,v) \in \mathcal{E}(v)}} \left\{ P + \{(x,r,v)\} \Big| P \in \hat{\mathcal{P}}^{(t-1)}_{u \rightsquigarrow x|q} \right\} \tag{8}$$

where $n^{(t)}_{uq} : 2^{\mathcal{V}} \mapsto 2^{\mathcal{V}}$ selects ending nodes of important paths of length $t$ from a set of nodes.

**Reasoning with A\* Algorithm**   Eqn. 8 iteratively computes the set of important paths $\hat{\mathcal{P}}_{u \leadsto v|q}$. In order to perform reasoning, we need to compute the representation $\boldsymbol{h}_q(u, v)$ based on the important paths, which can be achieved by an iterative process similar to Eqn. 8 (see Thm. A.4 in App. A)

$$\boldsymbol{h}_q^{(t)}(u,v) \leftarrow \boldsymbol{h}_q^{(0)}(u,v) \oplus \bigoplus_{\substack{x \in n_{uq}^{(t-1)}(\mathcal{V}) \\ (x,r,v) \in \mathcal{E}(v)}} \boldsymbol{h}_q^{(t-1)}(u,x) \otimes \boldsymbol{w}_q(x,r,v) \tag{9}$$

Eqn. 9 is the A\* iteration (Fig. 2(d)) for path-based reasoning. Note the A\* iteration uses the same boundary condition as Eqn. 2. Inspired by the classical A\* algorithm, we parameterize $n_{uq}^{(t)}(\mathcal{V})$ with a node priority function $s_{uq}^{(t)} : \mathcal{V} \mapsto [0, 1]$ and select top-$K$ nodes based on their priority. However, there does not exist an oracle for the priority function $s_{uq}^{(t)}(x)$. We will discuss how to learn the priority function $s_{uq}^{(t)}(x)$ in the following sections.

## 3.2   Path-based Reasoning with A\*Net

Both the performance and the efficiency of the A\* algorithm heavily rely on the heuristic function. While it is straightforward to use $L_1$ distance as the heuristic function for grid-world shortest path problems, it is not clear what a good priority function for knowledge graph reasoning is due to the complex relation semantics in knowledge graphs. Indeed, our experiments suggest that handcrafted priority functions largely hurt the performance of path-based methods (Tab. 6a). In this section, we discuss a neural priority function, which can be end-to-end trained by the reasoning task.

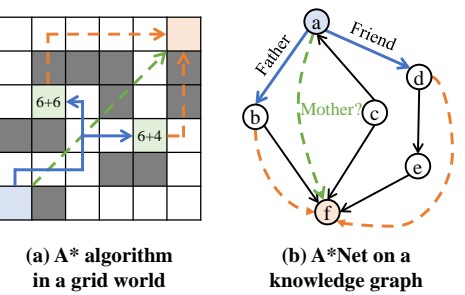

(a) A\* algorithm in a grid world          (b) A\*Net on a knowledge graph

Figure 4: **(a)** A\* algorithm computes the current distance $d(u, x)$ (blue), estimates the remaining distance $g(x, v)$ (orange), and prioritizes shorter paths. **(b)** A\*Net computes the current representations $\boldsymbol{h}_q^{(t)}(u, x)$ (blue), estimates the remaining representations $\boldsymbol{g}([\boldsymbol{h}_q^{(t)}(u, x), \boldsymbol{q}])$ (orange) based on the query $\boldsymbol{q}$ (green), and prioritizes paths more relevant to the query.

**Neural Priority Function**   To design the neural priority function $s_{uq}(x)$, we draw inspiration from the priority function in the A\* algorithm for shortest path problems (Eqn. 4). The priority function has two terms $d(u, x)$ and $g(x, v)$, where $d(u, x)$ is the current distance from node $u$ to $x$, and $g(x, v)$ estimates the remaining distance from node $x$ to $v$.

From a representation learning perspective, we need to learn a representation $\boldsymbol{s}_{uq}(x)$ to predict the priority score $s_{uq}(x)$ for each node $x$. Inspired by Eqn. 4, we use the current representation $\boldsymbol{h}_q^{(t)}(u, x)$ to represent $d^{(t)}(u, x)$. However, it is challenging to find a representation for $g^{(t)}(x, v)$, since we do not know the answer entity $v$ beforehand. Noticing that in the A\* algorithm, the target node $v$ can be expressed by the source node plus a displacement (Fig. 4(a)), we reparameterize the answer entity $v$ with the head entity $u$ and the query relation $q$ in A\*Net. By replacing $g^{(t)}(x, v)$ with another function $g^{(t)}(u, x, q)$, the representation $\boldsymbol{s}_{uq}(x)$ is parameterized as

$$\boldsymbol{s}_{uq}^{(t)}(x) = \boldsymbol{h}_q^{(t)}(u, x) \otimes \boldsymbol{g}([\boldsymbol{h}_q^{(t)}(u, x), \boldsymbol{q}]) \tag{10}$$

where $\boldsymbol{g}(\cdot)$ is a feed-forward network that outputs a vector representation and $[\cdot, \cdot]$ concatenates two representations. Intuitively, the learned representation $\boldsymbol{q}$ captures the semantic of query relation $q$, which serves the goal for answering query $(u, q, ?)$. The function $\boldsymbol{g}([\boldsymbol{h}_q^{(t)}(u, x), \boldsymbol{q}])$ compares the current representation $\boldsymbol{h}_q^{(t)}(u, x)$ with the goal $\boldsymbol{q}$ to estimate the remaining representation (Fig. 4(b)). If $\boldsymbol{h}_q^{(t)}(u, x)$ is close to $\boldsymbol{q}$, the remaining representation will be close to 0, and $x$ is likely to be close to the correct answer. The final priority score is predicted by

$$s_{uq}^{(t)}(x) = \sigma(f(\boldsymbol{s}_{uq}^{(t)}(x))) \tag{11}$$

where $f(\cdot)$ is a feed-forward network and $\sigma$ is the sigmoid function that maps the output to $[0, 1]$.

**Learning**  To learn the neural priority function, we incorporate it as a weight for each message in the A* iteration. For simplicity, let $\mathcal{X}^{(t)} = n_{uq}^{(t-1)}(\mathcal{V})$ be the nodes we try to propagate through at $t$-th iteration. We modify Eqn. 9 to be

$$\boldsymbol{h}_q^{(t)}(u,v) \leftarrow \boldsymbol{h}_q^{(0)}(u,v) \oplus \bigoplus_{\substack{x \in \mathcal{X}^{(t)} \\ (x,r,v) \in \mathcal{E}(v)}} s_{uq}^{(t-1)}(x) \left( \boldsymbol{h}_q^{(t-1)}(u,x) \otimes \boldsymbol{w}_q(x,r,v) \right) \qquad (12)$$

Eqn. 12 encourages the model to learn larger weights $s_{uq}^{(t)}(x)$ for nodes that are important for reasoning. In practice, as some nodes may have very large degrees, we further select top-$L$ edges from the neighborhood of $n_{uq}^{(t-1)}(\mathcal{V})$ (see App. B). A pseudo code of A*Net is illustrated in Alg. 1. Note the top-$K$ and top-$L$ functions are not differentiable.

Nevertheless, it is still too challenging to train the neural priority function, since we do not know the ground truth for important paths, and there is no direct supervision for the priority function. Our solution is to share the weights between the priority function and the predictor for the reasoning task. The intuition is that the reasoning task can be viewed as a weak supervision for the priority function. Recall that the goal of $s_{uq}^{(t)}(x)$ is to determine whether there exists an important path from $u$ to $x$ (Eqn. 8). In the reasoning task, any positive answer entity must be present on at least one important path, while negative answer entities are less likely to be on important paths. Our ablation experiment demonstrates that sharing weights improve the performance of neural priority function (Tab. 6b). Following [38], A*Net is trained to minimize the binary cross entropy loss over triplets

$$\mathcal{L} = -\log p(u,q,v) - \sum_{i=1}^{n} \frac{1}{n} \log(1 - p(u_i',q,v_i')) \qquad (13)$$

where $(u,q,v)$ is a positive sample and $\{(u_i',q,v_i')\}_{i=1}^{n}$ are negative samples. Each negative sample $(u_i,q,v_i)$ is generated by corrupting the head or the tail in a positive sample.

**Efficient Implementation with Padding-Free Operations**  Modern neural networks heavily rely on batched execution to unleash the parallel capacity of GPUs. While Alg. 1 is easy to implement for a single sample $(u,q,?)$, it is not trivial to batch A*Net for multiple samples. The challenge is that different samples may have very different sizes for nodes $\mathcal{V}^{(t)}$ and edges $\mathcal{E}^{(t)}$. A common approach is to pad the set of nodes or edges to a predefined constant, which would severely counteract the acceleration brought by A*Net.

Here we introduce padding-free $topk$ operation to avoid the overhead in batched execution. The key idea is to convert batched execution of different small samples into execution of a single large sample, which can be paralleled by existing operations in deep learning frameworks. For example, the batched execution of $topk([[1,3],[2,1,0]])$ can be converted into a multi-key sort problem over

---

**Algorithm 1** A*Net

**Input:** head entity $u$, query relation $q$, #iterations $T$
**Output:** $p(v|u,q)$ for all $v \in \mathcal{V}$
1: **for** $v \in \mathcal{V}$ **do**
2:     $\boldsymbol{h}_q^{(0)}(u,v) \leftarrow \mathbb{1}_q(u=v)$
3: **end for**
4: **for** $t \leftarrow 1$ to $T$ **do**
5:     $\mathcal{X}^{(t)} \leftarrow \text{TopK}(s_{uq}^{(t-1)}(x)|x \in \mathcal{V})$
6:     $\mathcal{E}^{(t)} \leftarrow \bigcup_{x \in \mathcal{X}^{(t)}} \mathcal{E}(x)$
7:     $\mathcal{E}^{(t)} \leftarrow \text{TopL}(s_{uq}^{(t-1)}(v)|(x,r,v) \in \mathcal{E}^{(t)})$
8:     $\mathcal{V}^{(t)} \leftarrow \bigcup_{(x,r,v) \in \mathcal{E}^{(t)}} \{v\}$
9:     **for** $v \in \mathcal{V}^{(t)}$ **do**
10:         Compute $\boldsymbol{h}_q^{(t)}(u,v)$ with Eqn. 12
11:         Compute priority $s_{uq}^{(t)}(v)$ with Eqn. 10, 11
12:     **end for**
13: **end for**
14: ▷ Share weights between $s_{uq}(v)$ and the predictor
15: **return** $s_{uq}^{(T)}(v)$ as $p(v|u,q)$ for all $v \in \mathcal{V}$

---

$[[0,1],[0,3],[1,2],[1,1],[1,0]]$, where the first key is the index of the sample in the batch and the second key is the original input. The multi-key sort is then implemented by composing stable single-key sort operations in deep learning frameworks. See App. C for details.

## 4 Experiments

We evaluate A*Net on standard transductive and inductive knowledge graph reasoning datasets, including a million-scale one ogbl-wikikg2. We conduct ablation studies to verify our design choices and visualize the important paths learned by the priority function in A*Net.

## 4.1 Experiment Setup

**Datasets & Evaluation**   We evaluate A*Net on 4 standard knowledge graphs, FB15k-237 [40], WN18RR [16], YAGO3-10 [30] and ogbl-wikikg2 [25]. For the transductive setting, we use the standard splits from their original works [40, 16]. For the inductive setting, we use the splits provided by [39], which contains 4 different versions for each dataset. As for evaluation, we use the standard filtered ranking protocol [6] for knowledge graph reasoning. Each triplet $(u, q, v)$ is ranked against all negative triplets $(u, q, v')$ or $(u', q, v)$ that are not present in the knowledge graph. We measure the performance with mean reciprocal rank (MRR) and HITS at K (H@K). Efficiency is measured by the average number of messages (#message) per step, wall time per epoch and memory cost. To plot the convergence curves for each model, we dump checkpoints during training with a high frequency, and evaluate the checkpoints later on the validation set. See more details in App. D.

**Implementation Details**   Our work is developed based on the open-source codebase of path-based reasoning with Bellman-Ford algorithm[4]. For a fair comparison with existing path-based methods, we follow the implementation of NBFNet [58] and parameterize $\bigoplus$ with principal neighborhood aggregation (PNA) [13] or sum aggregation, and parameterize $\bigotimes$ with the relation operation from DistMult [49], i.e., vector multiplication. The indicator function (Eqn. 2) $\mathbb{1}_q(u = v) = \mathbb{1}(u = v)\boldsymbol{q}$ is parameterized with a query embedding $\boldsymbol{q}$ for all datasets except ogbl-wikikg2, where we augment the indicator function with learnable embeddings based on a soft distance from $u$ to $v$ (see App. E for more details). The edge representation (Eqn. 12) $\boldsymbol{w}_q(x, r, v) = \boldsymbol{W}_r\boldsymbol{q} + \boldsymbol{b}_r$ is parameterized as a linear function over the query relation $q$ for all datasets except WN18RR, where we use a simple embedding $\boldsymbol{w}_q(x, r, v) = \boldsymbol{r}$. We use the same preprocessing steps as in [58], including augmenting each triplet with a flipped triplet, and dropping out query edges during training.

For the neural priority function, we have two hyperparameters: $K$ for the maximum number of nodes and $L$ for the maximum number of edges. To make hyperparameter tuning easier, we define maximum node ratio $\alpha = K/|\mathcal{V}|$ and maximum average degree ratio $\beta = L|\mathcal{V}|/K|\mathcal{E}|$, and tune the ratios for each dataset. The maximum edge ratio is determined by $\alpha\beta$. The other hyperparameters are kept the same as the values in [58]. We train A*Net with 4 Tesla A100 GPUs (40 GB), and select the best model based on validation performance. See App. E for more details.

**Baselines**   We compare A*Net against embedding methods, GNNs and path-based methods. The embedding methods are TransE [6], ComplEx [42], RotatE [38], HAKE [55], RotH [7], PairRE [8], ComplEx+Relation Prediction [12] and ConE [3]. The GNNs are RGCN [36], CompGCN [43] and GraIL [39]. The path-based methods are MINERVA [14], Multi-Hop [29], CURL [52], NeuralLP [50], DRUM [35], NBFNet [58] and RED-GNN [54]. Note that path-finding methods [14, 29, 52] that use reinforcement learning and assume sparse answers can only be evaluated on tail prediction. Training time of all baselines are measured based on their official open-source implementations, except that we use a more recent implementation[5] of TransE and ComplEx.

## 4.2 Main Results

Tab. 1 shows that A*Net outperforms all embedding methods and GNNs, and is on par with NBFNet on transductive knowledge graph reasoning. We also observe a similar trend of A*Net and NBFNet over path-finding methods on tail prediction (Tab. 2). Since path-finding methods select only one path with reinforcement learning, such results imply the advantage of aggregating multiple paths in A*Net. A*Net also converges faster than all the other methods (Fig. 5). Notably, unlike NBFNet that propagates through all nodes and edges, A*Net only propagates through 10% nodes and 10% edges on both datasets, which suggests that most nodes and edges are not important for path-based reasoning. Tab. 3 shows that A*Net reduces the number of messages by 14.1× and 42.9× compared to NBFNet on two datasets respectively. Note that the reduction in time and memory is less than the reduction in the number of messages, since A*Net operates on subgraphs with dynamic sizes and is harder to parallel than NBFNet on GPUs. We leave better parallel implementation as future work.

Tab. 4 shows the performance on ogbl-wikikg2, which has 2.5 million entities and 16 million triplets. While NBFNet faces out-of-memory (OOM) problem even for a batch size of 1, A*Net can perform reasoning by propagating through 0.2% nodes and 0.2% edges at each step. Surprisingly, even with

---

[4]`https://github.com/DeepGraphLearning/NBFNet`. MIT license.
[5]`https://github.com/DeepGraphLearning/KnowledgeGraphEmbedding`. MIT license.

Table 1: Performance on transductive knowledge graph reasoning. Results of embedding methods are from [3]. Results of GNNs and path-based methods are from [58]. Performance and efficiency on YAGO3-10 are in App. F.

| Method | FB15k-237 | | | | WN18RR | | | |
|--------|-----|------|------|-------|-----|------|------|-------|
| | MRR | H@1 | H@3 | H@10 | MRR | H@1 | H@3 | H@10 |
| TransE | 0.294 | - | - | 0.465 | 0.226 | - | 0.403 | 0.532 |
| RotatE | 0.338 | 0.241 | 0.375 | 0.533 | 0.476 | 0.428 | 0.492 | 0.571 |
| HAKE | 0.341 | 0.243 | 0.378 | 0.535 | 0.496 | 0.451 | 0.513 | 0.582 |
| RotH | 0.344 | 0.246 | 0.380 | 0.535 | 0.495 | 0.449 | 0.514 | 0.586 |
| ComplEx+RP | 0.388 | 0.298 | 0.425 | 0.568 | 0.488 | 0.443 | 0.505 | 0.578 |
| ConE | 0.345 | 0.247 | 0.381 | 0.540 | 0.496 | 0.453 | 0.515 | 0.579 |
| RGCN | 0.273 | 0.182 | 0.303 | 0.456 | 0.402 | 0.345 | 0.437 | 0.494 |
| CompGCN | 0.355 | 0.264 | 0.390 | 0.535 | 0.479 | 0.443 | 0.494 | 0.546 |
| NeuralLP | 0.240 | - | - | 0.362 | 0.435 | 0.371 | 0.434 | 0.566 |
| DRUM | 0.343 | 0.255 | 0.378 | 0.516 | 0.486 | 0.425 | 0.513 | 0.586 |
| NBFNet | **0.415** | **0.321** | **0.454** | **0.599** | **0.551** | **0.497** | **0.573** | **0.666** |
| RED-GNN | 0.374 | 0.283 | - | 0.558 | 0.533 | 0.485 | - | 0.624 |
| A*Net | **0.411** | **0.321** | **0.453** | 0.586 | **0.549** | **0.495** | **0.573** | **0.659** |

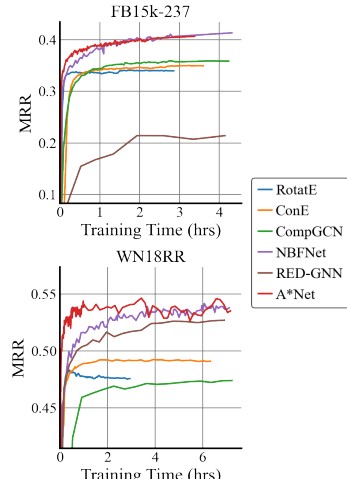

Figure 5: Validation MRR w.r.t. training time (1 A100 GPU).

Table 2: Tail prediction performance on transductive knowledge graphs. Results of compared methods are from [14, 29, 52].

| Method | FB15k-237 | | | | WN18RR | | | |
|--------|-----|------|------|-------|-----|------|------|-------|
| | MRR | H@1 | H@3 | H@10 | MRR | H@1 | H@3 | H@10 |
| MINERVA | 0.293 | 0.217 | 0.329 | 0.456 | 0.448 | 0.413 | 0.456 | 0.513 |
| Multi-Hop | 0.393 | 0.329 | - | 0.544 | 0.472 | 0.437 | - | 0.542 |
| CURL | 0.306 | 0.224 | 0.341 | 0.470 | 0.460 | 0.429 | 0.471 | 0.523 |
| NBFNet | **0.509** | **0.411** | **0.562** | **0.697** | **0.557** | 0.503 | 0.579 | **0.669** |
| A*Net | **0.505** | **0.410** | **0.556** | 0.687 | **0.557** | **0.504** | **0.580** | **0.666** |

Table 4: Performance on ogbl-wikikg2 (MRR). Results of compared methods are from [8, 12].

| Method | ogbl-wikikg2 | | |
|--------|------|-------|---------|
| | Test | Valid | #Params |
| TransE | 0.4256 | 0.4272 | 1,251 M |
| ComplEx | 0.4027 | 0.3759 | 1,251 M |
| RotatE | 0.4332 | 0.4353 | 1,251 M |
| PairRE | 0.5208 | 0.5423 | 500 M |
| ComplEx+RP | 0.6392 | 0.6561 | 250 M |
| NBFNet | OOM | OOM | OOM |
| A*Net | **0.6767** | **0.6851** | **6.83 M** |

Table 3: Efficiency on transductive knowledge graph reasoning.

| Method | FB15k-237 | | | WN18RR | | |
|--------|----------|---------|--------|----------|---------|--------|
| | #message | time | memory | #message | time | memory |
| NBFNet | 544,230 | 16.8 min | 19.1 GiB | 173,670 | 9.42 min | 26.4 GiB |
| A*Net | 38,610 | 8.07 min | 11.1 GiB | 4,049 | 1.39 min | 5.04 GiB |
| Improvement | 14.1× | 2.1× | 1.7× | 42.9× | 6.8× | 5.2× |

such sparse propagation, A*Net outperforms embedding methods and achieves a new state-of-the-art result. Moreover, the validation curve in Fig. 1 shows that A*Net converges significantly faster than embedding methods. Since A*Net only learns parameters for relations but not entities, it only uses 6.83 million parameters, which is 36.6× less than the best embedding method ComplEx+RP.

Tab. 5 shows the performance on inductive knowledge graph reasoning. A*Net is on par with NBFNet and significantly outperforms all the other methods. Note that embedding methods cannot deal with the inductive setting. Other metrics (H@1, H@10) and efficiency results are in App. F.

## 4.3 Ablation Studies

**Priority Function**   To verify the effectiveness of neural priority function, we compare it against three handcrafted priority functions: personalized PageRank (PPR), Degree and Random. PPR selects nodes with higher PPR scores w.r.t. the query head entity $u$. Degree selects nodes with larger degrees, while Random selects nodes uniformly. Tab. 6 shows that the neural priority function outperforms all three handcrafted priority functions, suggesting the necessity of learning a neural priority function.

**Sharing Weights**   As discussed in Sec. 3.2, we share the weights between the neural priority function and the reasoning predictor to help train the neural priority function. Tab. 6 compares A*Net trained with and without sharing weights. It can be observed that sharing weights is essential to training a good neural priority function in A*Net.

**Trade-off between Performance and Efficiency**   While A*Net matches the performance of NBFNet in less training time, one may further trade off performance and efficiency in A*Net by adjusting the ratios $\alpha$ and $\beta$. Fig. 6 plots curves of performance and speedup ratio w.r.t. different

Table 5: Performance on inductive knowledge graph reasoning (MRR). V1-v4 are 4 standard inductive splits. Results of compared methods are taken from [54]. $\alpha = 50\%$ and $\beta = 100\%$ for FB15k237. $\alpha = 5\%$ and $\beta = 100\%$ for WN18RR. More metrics and efficiency results are in App. F.

| Method | FB15k-237 | | | | WN18RR | | | |
| --- | --- | --- | --- | --- | --- | --- | --- | --- |
| | v1 | v2 | v3 | v4 | v1 | v2 | v3 | v4 |
| GraIL | 0.279 | 0.276 | 0.251 | 0.227 | 0.627 | 0.625 | 0.323 | 0.553 |
| NeuralLP | 0.325 | 0.389 | 0.400 | 0.396 | 0.649 | 0.635 | 0.361 | 0.628 |
| DRUM | 0.333 | 0.395 | 0.402 | 0.410 | 0.666 | 0.646 | 0.380 | 0.627 |
| NBFNet | 0.422 | **0.514** | **0.476** | 0.453 | **0.741** | **0.704** | **0.452** | 0.641 |
| RED-GNN | 0.369 | 0.469 | 0.445 | 0.442 | 0.701 | 0.690 | 0.427 | 0.651 |
| A*Net | **0.457** | 0.510 | **0.476** | **0.466** | 0.727 | **0.704** | 0.441 | **0.661** |

Table 6: Ablation studies of A*Net on transductive FB15k-237.

(a) Choices of priority function.

| Priority Function | FB15k-237 | | | |
| --- | --- | --- | --- | --- |
| | MRR | H@1 | H@3 | H@10 |
| PPR | 0.266 | 0.212 | 0.296 | 0.371 |
| Degree | 0.347 | 0.268 | 0.383 | 0.501 |
| Random | 0.378 | 0.288 | 0.413 | 0.556 |
| Neural | **0.411** | **0.321** | **0.453** | **0.586** |

(b) W/ or w/o sharing weights.

| Sharing Weights | FB15k-237 | | | |
| --- | --- | --- | --- | --- |
| | MRR | H@1 | H@3 | H@10 |
| No | 0.374 | 0.282 | 0.413 | 0.557 |
| Yes | **0.411** | **0.321** | **0.453** | **0.586** |

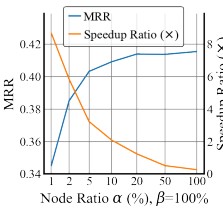
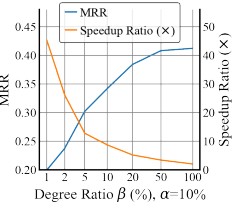

Figure 6: Performance and efficiency trade-off w.r.t. node ratio $\alpha$ and degree ratio $\beta$. Speedup ratio is relative to NBFNet.

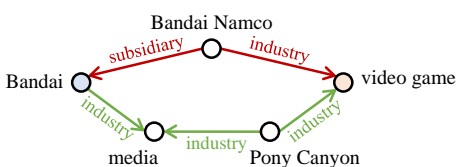

Figure 7: Visualization of important paths learned by the neural priority function in A*Net.

$\alpha$ and $\beta$. If we can accept a performance similar to embedding methods (e.g., ConE [3]), we can set either $\alpha$ to 1% or $\beta$ to 10%, resulting in 8.7× speedup compared to NBFNet.

### 4.4 Visualization of Learned Important Paths

We can extract the important paths from the neural priority function in A*Net for interpretation. For a given query $(u, q, ?)$ and a predicted entity $v$, we can use the node priority $s_{uq}^{(t)}(x)$ at each step to estimate the importance of a path. Empirically, the importance of a path $s_q(P)$ is estimated by

$$s_q(P) = \frac{1}{|P|} \sum_{t=1, P^{(t)}=(x,r,y)}^{|P|} \frac{s_{uq}^{(t-1)}(x)}{S_{uq}^{(t-1)}} \tag{14}$$

where $S_{uq}^{(t-1)} = \max_{x \in \mathcal{V}^{(t-1)}} s_{uq}^{(t-1)}(x)$ is a normalizer to normalize the priority score for each step $t$. To extract the important paths with large $s_q(P)$, we perform beam search over the priority function $s_{uq}^{(t-1)}(x)$ of each step. Fig. 7 shows the important paths learned by A*Net for a test sample in FB15k-237. Given the query *(Bandai, industry, ?)*, we can see both paths *Bandai* $\xleftarrow{subsidiary}$ *Bandai Namco* $\xrightarrow{industry}$ *video game* and *Bandai* $\xrightarrow{industry}$ *media* $\xleftarrow{industry}$ *Pony Canyon* $\xrightarrow{industry}$ *video game* are consistent with human cognition. More visualization results can be found in App. G.

## 5 Related Work

**Path-based Reasoning** Path-based methods use paths between entities for knowledge graph reasoning. Early methods like Path Ranking [28, 19] collect relational paths as symbolic features for classification. Path-RNN [31, 15] and PathCon [45] improve Path Ranking by learning the representations of paths with recurrent neural networks (RNN). However, these works operate on the full set of paths between two entities, which grows exponentially w.r.t. the path length. Typically, these methods can only be applied to paths with at most 3 edges.

To avoid the exhaustive search of paths, many methods learn to sample important paths for reasoning. DeepPath [47] and MINERVA [14] learn an agent to collect meaningful paths on the knowledge graph

through reinforcement learning. These methods are hard to train due to the extremely sparse rewards. Later works improve them by engineering the reward function [29] or the search strategy [37], using multiple agents for positive and negative paths [24] or for coarse- and fine-grained paths [52]. [11] and [33] use a variational formulation to learn a sparse prior for path sampling. Another category of methods utilizes the dynamic programming to search paths in a polynomial time. NeuralLP [50] and DRUM [35] use dynamic programming to learn linear combination of logic rules. All-Paths [41] adopts a Floyd-Warshall-like algorithm to learn path representations between all pairs of entities. Recently, NBFNet [58] and RED-GNN [54] leverage a Bellman-Ford-like algorithm to learn path representations from a single-source entity to all entities. While dynamic programming methods achieve state-of-the-art results among path-based methods, they need to perform message passing on the full knowledge graph. By comparison, our A*Net learns a priority function and only explores a subset of paths, which is more scalable than existing dynamic programming methods.

**Efficient Graph Neural Networks** Our work is also related to efficient graph neural networks, since both try to improve the scalability of graph neural networks (GNNs). Sampling methods [21, 9, 26, 51] reduce the cost of message passing by computing GNNs with a sampled subset of nodes and edges. Non-parametric GNNs [27, 46, 18, 10] decouple feature propagation from feature transformation, and reduce time complexity by preprocessing feature propagation. However, both sampling methods and non-parametric GNNs are designed for homogeneous graphs, and it is not straightforward to adapt them to knowledge graphs. On knowledge graphs, RS-GCN [17] learns to sample neighborhood with reinforcement learning. DPMPN [48] learns an attention to iteratively select nodes for message passing. SQALER [1] first predicts important path types based on the query, and then applies GNNs on the subgraph extracted by the predicted paths. Our A*Net shares the same goal with these methods, but learns a neural priority function to iteratively select important paths.

## 6 Discussion and Conclusion

**Limitation and Future Work** One limitation for A*Net is that we focus on algorithm design rather than system design. As a result, the improvement in time and memory cost is much less than the improvement in the number of messages (Tab. 3 and App. F). In the future, we will co-design the algorithm and the system to further improve the efficiency.

**Societal Impact** This work proposes a scalable model for path-based reasoning. On the positive side, it reduces the training and test time of reasoning models, which helps control carbon emission. On the negative side, reasoning models might be used in malicious activities, such as discovering sensitive relationship in anonymized data, which could be augmented by a more scalable model.

**Conclusion** We propose A*Net, a scalable path-based method, to solve knowledge graph reasoning by searching for important paths, which is guided by a neural priority function. Experiments on both transductive and inductive knowledge graphs verify the performance and efficiency of A*Net. Meanwhile, A*Net is the first path-based method that scales to million-scale knowledge graphs.

## Acknowledgement

This project is supported by Intel-MILA partnership program, the Natural Sciences and Engineering Research Council (NSERC) Discovery Grant, the Canada CIFAR AI Chair Program, collaboration grants between Microsoft Research and Mila, Samsung Electronics Co., Ltd., Amazon Faculty Research Award, Tencent AI Lab Rhino-Bird Gift Fund and a NRC Collaborative R&D Project (AI4D-CORE-06). This project was also partially funded by IVADO Fundamental Research Project grant PRF-2019-3583139727. The computation resource of this project is supported by Mila[6], Calcul Québec[7] and the Digital Research Alliance of Canada[8].

We would like to thank Zuobai Zhang, Jiarui Lu and Minghao Xu for helpful discussions and comments. We also appreciate all anonymous reviewers for their constructive suggestions.

---

[6] https://mila.quebec/
[7] https://www.calculquebec.ca/
[8] https://alliancecan.ca/

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
