# A  Path-based Reasoning with A* Algorithm

Here we prove the correctness of path-based reasoning with A* algorithm.

## A.1  Iterative Path Selection for Computing Important Paths

First, we prove that $\hat{\mathcal{P}}_{u \rightsquigarrow v|q}^{(t)}$ computed by Eqn. 6 and 7 equals to the set of important paths and paths that are different from important paths in the last hop.

**Theorem A.1.** *If $m_q(\mathcal{P}) : 2^{\mathcal{P}} \mapsto 2^{\mathcal{P}}$ can select all important paths from a set of paths $\mathcal{P}$, the set of paths $\hat{\mathcal{P}}_{u \rightsquigarrow v|q}^{(t)}$ computed by Eqn. 6 and 7 equals to the set of important paths and paths that are different from important paths in the last hop of length $t$.*

$$\hat{\mathcal{P}}_{u \rightsquigarrow v|q}^{(0)} \leftarrow \{(u, self\ loop, v)\}\ if\ u = v\ else\ \varnothing \tag{6}$$

$$\hat{\mathcal{P}}_{u \rightsquigarrow v|q}^{(t)} \leftarrow \bigcup_{\substack{x \in \mathcal{V} \\ (x,r,v) \in \mathcal{E}(v)}} \left\{ P + \{(x,r,v)\} \middle| P \in m_q(\hat{\mathcal{P}}_{u \rightsquigarrow x|q}^{(t-1)}) \right\} \tag{7}$$

*Proof.* We use $\mathcal{Q}_{u \rightsquigarrow v|q}^{(t)}$ to denote the set of important paths and paths that are different from important paths in the last hop of length $t$. For paths of length 0, we define them to be important as they should be the prefix of some important paths. Therefore, $\mathcal{Q}_{u \rightsquigarrow v|q}^{(0)} = \{(u, \text{self loop}, v)\}$ if $u = v$ else $\varnothing$. We use $P_{:-1}$ to denote the prefix of path $P$ without the last hop. The goal is to prove $\hat{\mathcal{P}}_{u \rightsquigarrow v|q}^{(t)} = \mathcal{Q}_{u \rightsquigarrow v|q}^{(t)}$.

First, we prove $\hat{\mathcal{P}}_{u \rightsquigarrow v|q}^{(t)} \subseteq \mathcal{Q}_{u \rightsquigarrow v|q}^{(t)}$. It is obvious that $\hat{\mathcal{P}}_{u \rightsquigarrow v|q}^{(0)} \subseteq \mathcal{Q}_{u \rightsquigarrow v|q}^{(0)}$. In the case of $t > 0$, $\forall P \in \hat{\mathcal{P}}_{u \rightsquigarrow v|q}^{(t)}$, we have $P_{:-1} \in m_q(\hat{\mathcal{P}}_{u \rightsquigarrow v|q}^{(t-1)})$ according to Eqn. 7. Therefore, $P \in \mathcal{Q}_{u \rightsquigarrow v|q}^{(t)}$.

Second, we prove $\mathcal{Q}_{u \rightsquigarrow v|q}^{(t)} \subseteq \hat{\mathcal{P}}_{u \rightsquigarrow v|q}^{(t)}$ by induction. For the base case $t = 0$, it is obvious that $\mathcal{Q}_{u \rightsquigarrow v|q}^{(0)} \subseteq \hat{\mathcal{P}}_{u \rightsquigarrow v|q}^{(0)}$. For the inductive case $t > 0$, $\forall Q \in \mathcal{Q}_{u \rightsquigarrow v|q}^{(t)}$, $Q_{:-1}$ is an important path of length $t - 1$ according to the definition of $\mathcal{Q}_{u \rightsquigarrow v|q}^{(t)}$. $Q_{:-1} \in m_q(\mathcal{Q}_{u \rightsquigarrow v|q}^{(t-1)}) \subseteq \mathcal{Q}_{u \rightsquigarrow v|q}^{(t-1)}$ according to the definition of $m_q(\cdot)$ and $\mathcal{Q}_{u \rightsquigarrow v|q}^{(t-1)}$. Based on the inductive assumption, we get $Q_{:-1} \in \hat{\mathcal{P}}_{u \rightsquigarrow v|q}^{(t-1)}$. Therefore, $Q \in \hat{\mathcal{P}}_{u \rightsquigarrow v|q}^{(t)}$ according to Eqn. 7. □

As a corollary of Thm. A.1, $\hat{\mathcal{P}}_{u \rightsquigarrow v|q}$ is a slightly larger superset of the important paths $P_{u \rightsquigarrow v|q}$.

**Corollary A.2.** *If the end nodes of important paths are uniformly distributed in the knowledge graph, the expected size of $\hat{\mathcal{P}}_{u \rightsquigarrow v|q}^{(t)}$ is $\left| \mathcal{P}_{u \rightsquigarrow v|q}^{(t)} \right| + \frac{|\mathcal{E}|}{|\mathcal{V}|} \left| \mathcal{P}_{u \rightsquigarrow v|q}^{(t-1)} \right|$.*

*Proof.* Thm. A.1 indicates that $\hat{\mathcal{P}}_{u \rightsquigarrow v|q}^{(t)}$ contains two types of paths: important paths and paths that are different from important paths in the last hop of length $t$. The number of the first type is $\left| \mathcal{P}_{u \rightsquigarrow v|q}^{(t)} \right|$. Each of the second type corresponds to an important path of length $t - 1$. From an inverse perspective, each important path of length $t - 1$ generates $d$ paths of the second type for $\hat{\mathcal{P}}_{u \rightsquigarrow v|q}^{(t)}$, where $d$ is the degree of the end node in the path. If the end nodes are uniformly distributed in the knowledge graph, we have $\mathbb{E}\left[ \hat{\mathcal{P}}_{u \rightsquigarrow v|q}^{(t)} \right] = \left| \mathcal{P}_{u \rightsquigarrow v|q}^{(t)} \right| + \frac{|\mathcal{E}|}{|\mathcal{V}|} \left| \mathcal{P}_{u \rightsquigarrow v|q}^{(t-1)} \right|$. For real-world knowledge graphs, $\frac{|\mathcal{E}|}{|\mathcal{V}|}$ is usually a small constant (e.g., $\leq 50$), and $\left| \hat{\mathcal{P}}_{u \rightsquigarrow v|q}^{(t)} \right|$ is slightly larger than $\left| \mathcal{P}_{u \rightsquigarrow v|q}^{(t)} \right|$ in terms of complexity. □

## A.2  From Iterative Path Selection to Iterative Node Selection

Second, we demonstrate that Eqn. 7 can be solved by Eqn. 8 if paths with the same length and the same stop node can be merged.

**Proposition A.3.** *If $m_q(\mathcal{P})$ selects paths only based on the length $t$, the start node $u$ and the end node $x$ of each path, by replacing $m_q(\mathcal{P})$ with $n_{uq}^{(t)}(\mathcal{V})$, $\hat{\mathcal{P}}_{u \rightsquigarrow v|q}^{(t)}$ can be computed as follows*

$$\hat{\mathcal{P}}_{u \rightsquigarrow v|q}^{(t)} \leftarrow \bigcup_{\substack{x \in n_{uq}^{(t-1)}(\mathcal{V}) \\ (x,r,v) \in \mathcal{E}(v)}} \left\{ P + \{(x,r,v)\} \Big| P \in \hat{\mathcal{P}}_{u \rightsquigarrow x|q}^{(t-1)} \right\} \tag{8}$$

This proposition is obvious. As a result of Prop. A.3, we merge paths by their length and stop nodes, which turns the exponential tree search to a polynomial dynamic programming algorithm.

### A.3 Reasoning with A* Algorithm

Finally, we prove that the A* iteration (Eqn. 9) covers all important paths for reasoning (Eqn. 5).

**Theorem A.4.** *If $n_{uq}^{(t)}(\mathcal{V}) : 2^{\mathcal{V}} \mapsto 2^{\mathcal{V}}$ can determine whether paths from $u$ to $x$ are important or not, and $\langle \oplus, \otimes \rangle$ forms a semiring [23], the representation $\boldsymbol{h}_q(u, v)$ for path-based reasoning can be computed by*

$$\boldsymbol{h}_q^{(t)}(u, v) \leftarrow \boldsymbol{h}_q^{(0)}(u, v) \oplus \bigoplus_{\substack{x \in n_{uq}^{(t-1)}(\mathcal{V}) \\ (x,r,v) \in \mathcal{E}(v)}} \boldsymbol{h}_q^{(t-1)}(u, x) \otimes \boldsymbol{w}_q(x, r, v) \tag{9}$$

*Proof.* In order to prove Thm. A.4, we first prove a lemma for the analytic form of $\boldsymbol{h}_q^{(t)}(u, v)$, and then show that $\lim_{t \to \infty} \boldsymbol{h}_q^{(t)}(u, v)$ converges to the goal of path-based reasoning.

**Lemma A.5.** *Under the same condition as Thm. A.4, the intermediate representation $\boldsymbol{h}_q^{(t)}(u, v)$ computed by Eqn. 2 and 9 aggregates all important paths within a length of $t$ edges, i.e.*

$$\boldsymbol{h}_q^{(t)}(u, v) = \bigoplus_{P \in \hat{\mathcal{P}}_{u \rightsquigarrow v|q}^{(\leq t)}} \bigotimes_{i=1}^{|P|} \boldsymbol{w}_q(e_i) \tag{15}$$

*where $\hat{\mathcal{P}}_{u \rightsquigarrow v|q}^{(\leq t)} = \bigcup_{k=0}^{t} \hat{\mathcal{P}}_{u \rightsquigarrow v|q}^{(k)}$.*

*Proof.* We prove Lem. A.5 by induction. Let $\textcircled{0}_q$ and $\textcircled{1}_q$ denote the identity elements of $\oplus$ and $\otimes$ respectively. We have $\mathbb{1}_q(u = v) = \textcircled{1}_q$ if $u = v$ else $\textcircled{0}_q$. Note paths of length 0 only contain self loops, and we define them as important paths, since they should be prefix of some important paths.

For the base case $t = 0$, we have $\boldsymbol{h}_q^{(0)}(u, u) = \textcircled{1}_q = \bigoplus_{P \in \mathcal{P}_{u \rightsquigarrow u|q}:|P| \leq 0} \bigotimes_{i=1}^{|P|} \boldsymbol{w}_q(e_i)$ since the only path from $u$ to $u$ is the self loop, which has the representation $\textcircled{1}_q$. For $u \neq v$, we have $\boldsymbol{h}_q^{(0)}(u, v) = \textcircled{0}_q = \bigoplus_{P \in \mathcal{P}_{u \rightsquigarrow v|q}:|P| \leq 0} \bigotimes_{i=1}^{|P|} \boldsymbol{w}_q(e_i)$ since there is no important path from $u$ to $v$ within length 0.

For the inductive case $t > 0$, we have

$$\boldsymbol{h}_q^{(t)}(u, v) = \boldsymbol{h}_q^{(0)}(u, v) \oplus \bigoplus_{\substack{x \in n_{uq}^{(t-1)}(\mathcal{V}) \\ (x,r,v) \in \mathcal{E}(v)}} \boldsymbol{h}_q^{(t-1)}(u, x) \otimes \boldsymbol{w}_q(x, r, v) \tag{16}$$

$$= \boldsymbol{h}_q^{(0)}(u, v) \oplus \bigoplus_{\substack{x \in n_{uq}^{(t-1)}(\mathcal{V}) \\ (x,r,v) \in \mathcal{E}(v)}} \left( \bigoplus_{P \in \hat{\mathcal{P}}_{u \rightsquigarrow v|q}^{(\leq t-1)}} \bigotimes_{i=1}^{|P|} \boldsymbol{w}_q(e_i) \right) \otimes \boldsymbol{w}_q(x, r, v) \tag{17}$$

$$= \boldsymbol{h}_q^{(0)}(u, v) \oplus \bigoplus_{\substack{x \in n_{uq}^{(t-1)}(\mathcal{V}) \\ (x,r,v) \in \mathcal{E}(v)}} \left[ \bigoplus_{P \in \hat{\mathcal{P}}_{u \rightsquigarrow v|q}^{(\leq t-1)}} \left( \bigotimes_{i=1}^{|P|} \boldsymbol{w}_q(e_i) \right) \otimes \boldsymbol{w}_q(x, r, v) \right] \tag{18}$$

$$= \left( \bigoplus_{P \in \hat{\mathcal{P}}_{u \rightsquigarrow v|q}^{(0)}} \bigotimes_{i=1}^{|P|} \boldsymbol{w}_q(e_i) \right) \oplus \left( \bigoplus_{P \in \hat{\mathcal{P}}_{u \rightsquigarrow v|q}^{(\leq t)} \backslash \hat{\mathcal{P}}_{u \rightsquigarrow v|q}^{(0)}} \bigotimes_{i=1}^{|P|} \boldsymbol{w}_q(e_i) \right) \tag{19}$$

$$= \bigoplus_{P \in \hat{\mathcal{P}}_{u \rightsquigarrow v|q}^{(\leq t)}} \bigotimes_{i=1}^{|P|} \boldsymbol{w}_q(e_i), \tag{20}$$

where Eqn. 17 uses the inductive assumption, Eqn. 18 relies on the distributive property of $\otimes$ over $\oplus$, and Eqn. 19 uses Prop. A.3. In the above equations, $\bigotimes$ and $\otimes$ are always applied before $\bigoplus$ and $\oplus$. $\qquad\square$

Since $\mathcal{P}_{u \rightsquigarrow v|q}^{(t)} \subseteq \hat{\mathcal{P}}_{u \rightsquigarrow v|q}^{(t)}$, we have $\mathcal{P}_{u \rightsquigarrow v|q} \subseteq \hat{\mathcal{P}}_{u \rightsquigarrow v|q} \subseteq \mathcal{P}_{u \rightsquigarrow v}$. Based on Lem. A.5 and Eqn. 5, it is obvious to see that

$$\lim_{t \to \infty} \boldsymbol{h}_q^{(t)}(u, v) = \bigoplus_{P \in \hat{\mathcal{P}}_{u \rightsquigarrow v|q}} \boldsymbol{h}_q(P) \approx \bigoplus_{P \in \mathcal{P}_{u \rightsquigarrow v}} \boldsymbol{h}_q(P) = \boldsymbol{h}_q(u, v) \tag{21}$$

Therefore, Thm. A.4 holds. $\qquad\square$

## B  Additional Edge Selection Step in A*Net

As demonstrated in Sec. 3.2, A*Net selects top-$K$ nodes according to the current priority function, and computes the A* iteration

$$\boldsymbol{h}_q^{(t)}(u, v) \leftarrow \boldsymbol{h}_q^{(0)}(u, v) \oplus \bigoplus_{\substack{x \in \mathcal{X}^{(t)} \\ (x, r, v) \in \mathcal{E}(v)}} s_{uq}^{(t-1)}(x) \left( \boldsymbol{h}_q^{(t-1)}(u, x) \otimes \boldsymbol{w}_q(x, r, v) \right) \tag{12}$$

However, even if we choose a small $K$, Eqn. 12 may still propagate the messages to many nodes in the knowledge graph, resulting in a high computation cost. This is because some nodes in the knowledge graph may have very large degrees, e.g., the entity *Human* is connected to every person in the knowledge graph. In fact, it is not necessary to propagate the messages to every neighbor of a node, especially if the node has a large degree. Based on this observation, we propose to further select top-$L$ edges from the neighborhood of $\mathcal{X}^{(t)}$ to create $\mathcal{E}^{(t)}$

$$\mathcal{E}^{(t)} \leftarrow \text{TopL}(s_{uq}^{(t-1)}(v)|x \in \mathcal{X}^{(t)}, (x, r, v) \in \mathcal{E}(x)) \tag{22}$$

where each edge is picked according to the priority of node $v$, i.e., the tail node of an edge. By doing so, we reuse the neural priority function and avoid introducing any additional priority function. The intuition of Eqn. 22 is that if an edge $(x, r, v)$ goes to a node with a higher priority, it is likely we are propagating towards the answer entities. With the selected edges $\mathcal{E}^{(t)}$, the A* iteration becomes

$$\boldsymbol{h}_q^{(t)}(u, v) \leftarrow \boldsymbol{h}_q^{(0)}(u, v) \oplus \bigoplus_{\substack{x \in \mathcal{X}^{(t)} \\ (x, r, v) \in \mathcal{E}^{(t)}(v)}} s_{uq}^{(t-1)}(x) \left( \boldsymbol{h}_q^{(t-1)}(u, x) \otimes \boldsymbol{w}_q(x, r, v) \right) \tag{23}$$

which is also the implementation in Alg. 1.

## C  Padding-Free Operations

In A*Net, different training samples may have very different sizes for the selected nodes $\mathcal{V}^{(t)}$ and $\mathcal{E}^{(t)}$. To avoid the additional computation over padding in conventional batched execution, we introduce padding-free operations, which operates on the concatenation of samples without any padding.

Specifically, padding-free operations construct IDs for each sample in the batch, such that we can distinguish different samples when we apply operations to the whole batch. As showed in Fig. 8, for padding-free *topk*, we pair the inputs with their sample IDs, and cast the problem as a multi-key sort over the whole batch. The multi-key sort is implemented by two calls to standard stable sort

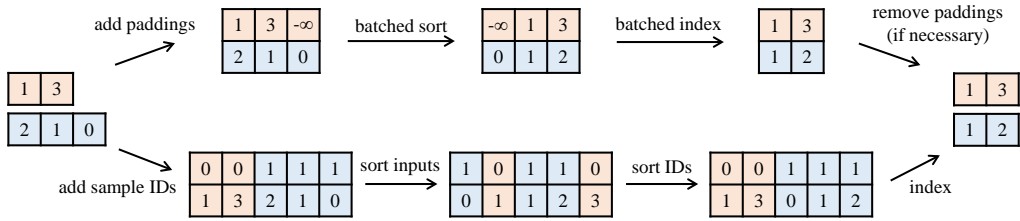

Figure 8: Comparison between padding-based *topk* (up) and padding-free *topk* (down) for $K = 2$. Padding-based operations first add paddings to create a padded tensor for batched operations, and then remove the paddings. Padding-free operations pair the inputs with their sample IDs (showed in colors), and then apply single-sample operations over the whole batch.

operations sequentially. We then apply indexing operations and remove the sample IDs to get the desired output. Alg. 2 provides the pseudo code for padding-free *topk* in PyTorch.

---

**Algorithm 2** Padding-free implementation of *topk* in PyTorch

---

**Input:** Input values of each sample `inputs`, size of each sample `sizes`, K
**Output:** TopK values of each sample, indices of topk values

```
1  # the sample id of each element
2  sample_ids = torch.arange(batch_size).repeat_interleave(sizes)
3  # multi-key sort of (sample_ids, inputs)
4  indices = inputs.argsort()
5  indices = sample_ids[indices].argsort(stable=True)
6  sorteds = inputs[indices]
7  # take top-k values of each sample
8  ranges = torch.arange(K).repeat(batch_size)
9  ranges = ranges + sizes.cumsum(0).repeat_interleave(K) - K
10 return sorteds[ranges], indices[ranges]
```

---

# D   Datasets & Evaluation

Dataset statistics for transductive and inductive knowledge graph reasoning is summarized in Tab. 7 and 8 respectively. For the transductive setting, given a query head (or tail) and a query relation, we rank each answer tail (or head) entity against all negative entities. For the inductive setting, we follow [54] and rank each each answer tail (or head) entity against all negative entities, rather than 50 randomly sampled negative entities in [39]. We report the mean reciprocal rank (MRR) and HITS at K (H@K) of the rankings.

Table 7: Dataset statistics for transductive knowledge graph reasoning.

| Dataset | #Relation | #Entity | #Triplet | | |
|---|---|---|---|---|---|
| | | | #Train | #Valid | #Test |
| FB15k-237 | 237 | 14,541 | 272,115 | 17,535 | 20,466 |
| WN18RR | 11 | 40,943 | 86,835 | 3,034 | 3,134 |
| YAGO3-10 | 37 | 123,182 | 1,079,040 | 5000 | 5000 |
| ogbl-wikikg2 | 535 | 2,500,604 | 16,109,182 | 429,456 | 598,543 |

As for efficiency evaluation, we compute the number of messages (#message) per step, wall time per epoch and memory cost. The number of messages is averaged over all samples and steps

$$\#\text{message} = \mathbb{E}_{(u,q,v)\in\mathcal{E}}\mathbb{E}_t \left| \mathcal{E}^{(t)} \right| \tag{24}$$

The wall time per epoch is defined as the average time to complete a single training epoch. We measure the wall time based on 10 epochs. The memory cost is measured by the function `torch.cuda.max_memory_allocated()` in PyTorch.

Table 8: Dataset statistics for inductive knowledge graph reasoning.

| Dataset | | #Relation | Train | | | Validation | | | Test | | |
|---------|----|-----------|---------|---------|---------|---------|---------|---------|---------|---------|---------|
| | | | #Entity | #Query | #Fact | #Entity | #Query | #Fact | #Entity | #Query | #Fact |
| FB15k-237 | v1 | 180 | 1,594 | 4,245 | 4,245 | 1,594 | 489 | 4,245 | 1,093 | 205 | 1,993 |
| | v2 | 200 | 2,608 | 9,739 | 9,739 | 2,608 | 1,166 | 9,739 | 1,660 | 478 | 4,145 |
| | v3 | 215 | 3,668 | 17,986 | 17,986 | 3,668 | 2,194 | 17,986 | 2,501 | 865 | 7,406 |
| | v4 | 219 | 4,707 | 27,203 | 27,203 | 4,707 | 3,352 | 27,203 | 3,051 | 1,424 | 11,714 |
| WN18RR | v1 | 9 | 2,746 | 5,410 | 5,410 | 2,746 | 630 | 5,410 | 922 | 188 | 1,618 |
| | v2 | 10 | 6,954 | 15,262 | 15,262 | 6,954 | 1,838 | 15,262 | 2,757 | 441 | 4,011 |
| | v3 | 11 | 12,078 | 25,901 | 25,901 | 12,078 | 3,097 | 25,901 | 5,084 | 605 | 6,327 |
| | v4 | 9 | 3,861 | 7,940 | 7,940 | 3,861 | 934 | 7,940 | 7,084 | 1,429 | 12,334 |

# E  Implementation Details

Our work is based on the open-source codebase of path-based reasoning with Bellman-Ford algorithm[9]. Tab. 9 lists the hyperparameters for A*Net on all datasets and in both transductive and inductive settings. For the inductive setting, we use the same set of hyperparameters for all 4 splits of each dataset.

Table 9: Hyperparameter configurations of A*Net on all datasets. For FB15k-237, WN18RR and YAGO3-10, we use the same hyperparameters as NBFNet [58], except for the neural priority function introduced in A*Net. There is no publicly available hyperparameters of NBFNet on ogbl-wikikg2.

| Hyperparameter | | FB15k-237 | | WN18RR | | YAGO3-10 | ogbl-wikikg2 |
|----------------|----|-----------|----------|--------|----------|----------|--------------|
| | | transductive | inductive | transductive | inductive | transductive | transductive |
| Message Passing | #step ($T$) | 6 | 6 | 6 | 6 | 6 | 6 |
| | hidden dim. | 32 | 32 | 32 | 32 | 32 | 32 |
| | message | DistMult | DistMult | DistMult | DistMult | DistMult | DistMult |
| | aggregation | PNA | sum | PNA | sum | PNA | sum |
| Priority Function | $g(\cdot)$ #layer | 1 | 1 | 1 | 1 | 1 | 1 |
| | $f(\cdot)$ #layer | 2 | 2 | 2 | 2 | 2 | 2 |
| | hidden dim. | 64 | 64 | 64 | 64 | 64 | 64 |
| | node ratio $\alpha$ | 10% | 50% | 10% | 5% | 10% | 0.2% |
| | degree ratio $\beta$ | 100% | 100% | 100% | 100% | 100% | 100% |
| Learning | optimizer | Adam | Adam | Adam | Adam | Adam | Adam |
| | batch size | 256 | 256 | 256 | 256 | 40 | 128 |
| | learning rate | 5e-3 | 5e-3 | 5e-3 | 5e-3 | 5e-3 | 5e-3 |
| | #epoch | 20 | 20 | 20 | 20 | 0.4 | 0.2 |
| | adv. temperature | 0.5 | 0.5 | 1 | 1 | 0.5 | 0.5 |
| | #negative | 32 | 32 | 32 | 32 | 32 | 1,048,576 |

**Neural Parameterization**  For a fair comparison with existing path-based methods, we follow NBFNet [58] and parameterize $\bigoplus$ with principal neighborhood aggregation (PNA), which is a permutation-invariant function over a set of elements. We parameterize $\bigotimes$ with the relation operation from DistMult [49], i.e., vector multiplication. Note that PNA relies on the degree information of each node to perform aggregation. We observe that PNA does not generalize well when degrees are dynamically determined by the priority function. Therefore, we precompute the degree for each node on the full graph, and use them in PNA no matter how many nodes and edges are selected by the priority function.

Following NBFNet [58], we parameterize the indicator function as $\mathbb{1}_q(u = v) = \mathbb{1}(u = v)\boldsymbol{q}$. Intuitively, this produces a boundary condition of zero vectors except for the head entity $u$, which is labeled with the query embedding $q$. For ogbl-wikikg2, instead of using a boundary condition of mostly zeros, we find it is better to incorporate distance information in the boundary condition. To this end, we use the personalized PageRank score $p_{u,v}$ from $u$ to $v$ as a soft distance metric, and parameterize the indicator function as $\mathbb{1}_q(u = v) = \mathbb{1}(u = v)\boldsymbol{q} + \mathbb{1}(u \neq v)\boldsymbol{p}_{u,v}$, where $\boldsymbol{p}_{u,v}$ is an embedding learned based on discretized value of $p_{u,v}$.

**Data Augmentation**  We follow the data augmentation steps of NBFNet [58]. For each triplet $(x, r, y)$, we add an inverse triplet $(y, r^{-1}, x)$ to the knowledge graph, so that A*Net can propagate in

---

[9]`https://github.com/DeepGraphLearning/NBFNet`. MIT license.

both directions. Each triplet and its inverse may have different priority and are picked independently in the edge selection step. Since test queries are always missing in the graph, we remove the edges of training queries during training to prevent the model from copying the input.

## F More Experiment Results

Tab. 10 shows the performance and efficiency results on YAGO3-10. We observe that A*Net achieves compatible performance with NBFNet, while reducing the number of messages by 16.0×. A*Net also reduces the time and memory of NBFNet by 2.5× and 2.0× respectively.

Tab. 11 provides all metrics of the performance on inductive knowledge graph reasoning. It can be observed that A*Net consistently outperforms all compared methods except NBFNet. A*Net achieves competitive performance compared to NBFNet, despite the fact that A*Net reduces the number of messages, wall time and memory on both datasets and all splits (Tab. 12).

Table 10: Performance and efficiency on YAGO3-10. Results of compared methods are from [38].

(a) Performance results.

| Method | YAGO3-10 | | | |
|---|---|---|---|---|
| | MRR | H@1 | H@3 | H@10 |
| DistMult | 0.34 | 0.24 | 0.38 | 0.54 |
| ComplEx | 0.36 | 0.26 | 0.40 | 0.55 |
| RotatE | 0.495 | 0.402 | 0.550 | 0.670 |
| NFBNet | **0.563** | **0.480** | **0.612** | **0.708** |
| A*Net | **0.556** | 0.470 | **0.611** | **0.707** |

(b) Efficiency results.

| Method | YAGO3-10 | | |
|---|---|---|---|
| | #message | time | memory |
| NBFNet | 2,158,080 | 51.3 min | 26.1 GiB |
| A*Net | 134,793 | 20.8 min | 13.1 GiB |
| Improvement | 16.0× | 2.5× | 2.0× |

Table 11: Performance on inductive knowledge graph reasoning. V1-v4 refer to the 4 standard splits.

| Method | v1 | | | v2 | | | v3 | | | v4 | | |
|---|---|---|---|---|---|---|---|---|---|---|---|---|
| | MRR | H@1 | H@10 | MRR | H@1 | H@10 | MRR | H@1 | H@10 | MRR | H@1 | H@10 |
| **FB15k-237** | | | | | | | | | | | | |
| GraIL | 0.279 | 0.205 | 0.429 | 0.276 | 0.202 | 0.424 | 0.251 | 0.165 | 0.424 | 0.227 | 0.143 | 0.389 |
| NeuralLP | 0.325 | 0.243 | 0.468 | 0.389 | 0.286 | 0.586 | 0.400 | 0.309 | 0.571 | 0.396 | 0.289 | 0.593 |
| DRUM | 0.333 | 0.247 | 0.474 | 0.395 | 0.284 | 0.595 | 0.402 | 0.308 | 0.571 | 0.410 | 0.309 | 0.593 |
| NBFNet | 0.422 | 0.335 | 0.574 | **0.514** | **0.421** | **0.685** | **0.476** | **0.384** | **0.637** | 0.453 | **0.360** | 0.627 |
| RED-GNN | 0.369 | 0.302 | 0.483 | 0.469 | 0.381 | 0.629 | 0.445 | 0.351 | 0.603 | 0.442 | 0.340 | 0.621 |
| A*Net | **0.457** | **0.381** | **0.589** | 0.510 | 0.419 | 0.672 | **0.476** | **0.389** | 0.629 | **0.466** | **0.365** | **0.645** |
| **WN18RR** | | | | | | | | | | | | |
| GraIL | 0.627 | 0.554 | 0.760 | 0.625 | 0.542 | 0.776 | 0.323 | 0.278 | 0.409 | 0.553 | 0.443 | 0.687 |
| NeuralLP | 0.649 | 0.592 | 0.772 | 0.635 | 0.575 | 0.749 | 0.361 | 0.304 | 0.476 | 0.628 | 0.583 | 0.706 |
| DRUM | 0.666 | 0.613 | 0.777 | 0.646 | 0.595 | 0.747 | 0.380 | 0.330 | 0.477 | 0.627 | 0.586 | 0.702 |
| NBFNet | **0.741** | **0.695** | **0.826** | **0.704** | **0.651** | 0.798 | **0.452** | **0.392** | **0.568** | 0.641 | 0.608 | 0.694 |
| RED-GNN | 0.701 | 0.653 | 0.799 | 0.690 | 0.633 | 0.780 | 0.427 | 0.368 | 0.524 | 0.651 | 0.606 | 0.721 |
| A*Net | 0.727 | 0.682 | 0.810 | **0.704** | 0.649 | **0.803** | 0.441 | **0.386** | 0.544 | **0.661** | **0.616** | **0.743** |

Table 12: Efficiency on inductive knowledge graph reasoning. V1-v4 refer to the 4 standard splits.

| Method | v1 | | | v2 | | | v3 | | | v4 | | |
|---|---|---|---|---|---|---|---|---|---|---|---|---|
| | #msg. | time | memory | #msg. | time | memory | #msg. | time | memory | #msg. | time | memory |
| **FB15k-237** | | | | | | | | | | | | |
| NBFNet | 8,490 | 4.50 s | 2.79 GiB | 19,478 | 11.3 s | 4.49 GiB | 35,972 | 27.2 s | 6.28 GiB | 54,406 | 50.1 s | 7.99 GiB |
| A*Net | 2,644 | 3.40 s | 0.97 GiB | 6,316 | 8.90 s | 1.60 GiB | 12,153 | 18.9 s | 2.31 GiB | 18,501 | 33.7 s | 3.05 GiB |
| Improvement | 3.2× | 1.3× | 2.9× | 3.1× | 1.3 × | 2.8× | 3.0× | 1.4× | 2.7× | 2.9× | 1.5× | 2.6× |
| **WN18RR** | | | | | | | | | | | | |
| NBFNet | 10,820 | 8.80 s | 1.79 GiB | 30,524 | 30.9 s | 4.48 GiB | 51,802 | 78.6 s | 7.75 GiB | 7,940 | 13.6 s | 2.49 GiB |
| A*Net | 210 | 2.85 s | 0.11 GiB | 478 | 8.65 s | 0.26 GiB | 704 | 13.2 s | 0.41 GiB | 279 | 4.20 s | 0.14 GiB |
| Improvement | 51.8× | 3.1× | 16.3× | 63.9× | 3.6× | 17.2× | 73.6× | 6.0× | 18.9× | 28.5× | 3.2× | 17.8× |

# G More Visualization of Learned Important Paths

Fig. 9 visualizes learned important paths on different samples. All the samples are picked from the test set of transductive FB15k-237.

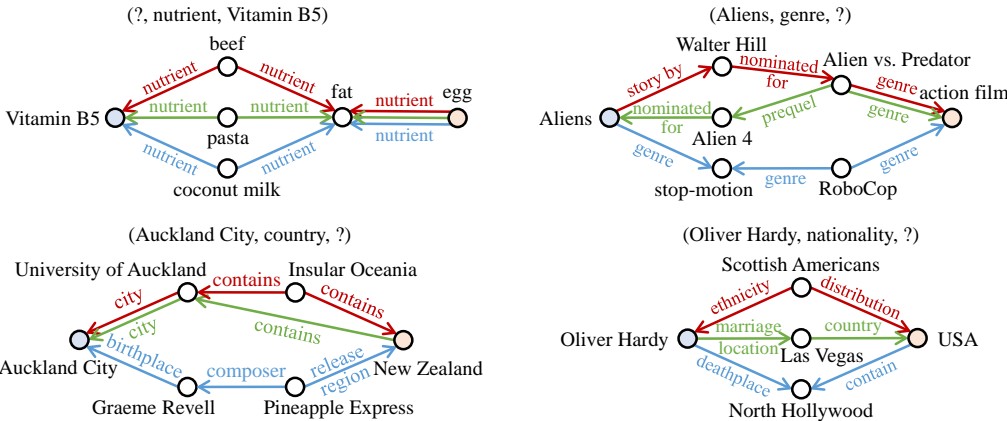

Figure 9: Visualization of important paths in A*Net on different test samples. Each important path is highlighted by a separate color.