# OpenReview forum: "A*Net: A Scalable Path-based Reasoning Approach for Knowledge Graphs"
_NeurIPS.cc/2023/Conference — NeurIPS 2023 poster_

### Official Review · Reviewer_ALvG · 2023-07-05

**Soundness:** 2 fair
**Presentation:** 3 good
**Contribution:** 3 good
**Rating:** 6
**Confidence:** 4

**Summary:**

The paper introduces ANet, a scalable path-based approach for reasoning on extensive knowledge graphs (KGs). In contrast to embedding techniques, path-based methods exhibit inductive capabilities but encounter challenges in terms of scalability due to the exponential growth of paths. ANet addresses this issue by incorporating a priority function, inspired by the A\* algorithm for shortest-path problems, which enables the selection of crucial nodes and edges during each iteration. This novel approach effectively reduces the time and memory requirements for both training and inference processes.

**Strengths:**

S1: This paper proposes an efficient GNN called A\*Net for link prediction with good scalability.

S2: A\*Net shows impressive results on various KGs.

**Weaknesses:**

W1: Although the method proposed in this article has better scalability, the contributions from theoretical perspectives are incremental compared to NBFNet.

W2: The introduction of the parameter sharing between the priority function and predictor is somewhat unclear, and the reason why the reasoning task can be regarded as weak supervision for the priority function is not well explained.

**Questions:**

Q1: The priority function in A\*Net is similar to the attention used in RED-GNN except that A\*Net selects the nodes and edges according to the attention score. In the case where memory allows, how does the performance of A\* Net change when Top operation is disabled in Algorithm 1 (line 5 & line 7)?

Q2: If some nodes and edges are discarded in the early phase of model training, it may introduce incorrect inductive biases and prevent the model from training effectively. How do you address this issue to avoid such problems or why is this not an issue in A\*Net?

**Limitations:**

See **Weaknesses** and **Questions**.

---

> ### Author Rebuttal · Authors · 2023-08-08
>
> Thanks for your comments. Here is our response to your concerns.
>
> **W1: Although the method proposed in this article has better scalability, the contributions from theoretical perspectives are incremental compared to NBFNet.**
>
> A1: The major contribution of this paper is scalability, not the theoretical insights. Previous path-based methods like NBFNet can only deal with graphs of tens of thousands of nodes. Our A\*Net scales path-based methods to ogbl-wikikg2, a dataset containing 2.5 million entities and 16 million triplets, which is **2 magnitudes larger than what NBFNet can solve**. Note that ogbl-wikikg2 has been previously dominant by embedding methods, and none of the embedding methods are inductive or interpretable like A\*Net. Therefore, we believe this contribution is very important to the knowledge graph community and may potentially change future research directions as new abilities may emerge with scalability.
>
> The theoretical design insights from the A\* algorithm are a minor contribution compared to our empirical achievements. However, we should emphasize that the design insights led us to develop A\*Net without significant performance loss. As showed in Tab. 6(a), naive solutions like personalized PageRank or node degree sacrifice the performance when they prune the paths.
>
> **W2: The introduction of the parameter sharing between the priority function and predictor is somewhat unclear, and the reason why the reasoning task can be regarded as weak supervision for the priority function is not well explained.**
>
> A2: The priority function is $s_{uq}^{(t)}(x) = \sigma(f(\mathbf{s}_{uq}^{(t)}(x))$
>
> where $\mathbf{s}_{uq}^{(t)}(x)$ is the representation computed by Eqn. 10 and $f(\cdot)$ is a feed-forward network.
>
> The predictor function is $p(v|u,q) = \sigma(f'(\mathbf{s}_{uq}^{(T)}(v)))$
>
> where $\mathbf{s}_{uq}^{(T)}(v)$ is the representation from the last layer and $f'(\cdot)$ is a feed-forward network. Since two functions have similar formulas, we share the parameters between $f(\cdot)$ and $f'(\cdot)$.
>
> For the priority function, an ideal supervision should assign high scores to nodes on the important paths, and low scores to nodes that are not on the important paths. We notice that the reasoning task assigns labels of 1 for true answer nodes, and labels of 0 for false answer nodes. Since true answer nodes are always on the important paths, and false answer nodes are less likely to be on the important paths, the supervision from the reasoning task is correlated to the supervision we need for the priority function. Therefore, we share the weights between these two functions, and hope that a well-trained predictor can help the priority function to converge when the whole model is trained end-to-end. We illustrated this idea in Line 188-191 in the paper.
>
> **Q1: In the case where memory allows, how does the performance of A\* Net change when the topk operation is disabled in Algorithm 1?**
>
> A3: Our ablation study (Fig. 6) suggests that the performance of A\*Net hits a plateau when we select more nodes or edges than a certain threshold. We run experiments with the topk operations disabled, which makes A\*Net almost identical to NBFNet. Here are the results. Generally, the performance doesn’t increase and is upper bounded by NBFNet.
>
> |FB15k-237|MRR|H@1|H@3|H@10|
> |---|---|---|---|---|
> |A\*Net (10% nodes, 10% edges)|0.411|0.321|0.453|0.586|
> |A\*Net (no pruning)|0.407|0.314|0.445|0.590|
> |NBFNet|0.415|0.321|0.454|0.599|
>
> **Q2: If some nodes and edges are discarded in the early phase of model training, it may introduce incorrect inductive biases and prevent the model from training effectively. How do you address this issue to avoid such problems or why is this not an issue in A\*Net?**
>
> A4: That’s a valid concern. Empirically, this is not an issue for A\*Net, as A\*Net achieves competitive performance compared to its unpruned version, NBFNet. We conjecture the reason is that A\*Net converges in a curriculum learning fashion with the help of the weight sharing trick. Since every edge in the training graph is a training sample, we automatically have a curriculum for training path-based methods. That is, the training samples cover consecutive distances from 1 to some finite value. As long as we start from some reasonable ratios of nodes and edges, the predictor function can converge on answers of distance 1. Through the weight sharing trick, this helps the priority function to find important nodes of distance 1, and reach answers of distance 2. Then the predictor function converges on answers of distance 2. Eventually, this procedure enables A\*Net to converge on answers of all distances.

---

> > ### Comment · Reviewer_ALvG · 2023-08-18
> >
> > Most of my concerns are addressed by rebuttal, so I raise my score to 6. Your reply to Q2 is interesting and I am curious about if there are any experimental results supporting your conjecturing.

---

> > > ### Author Response · Authors · 2023-08-20
> > > **Discussion**
> > >
> > > Thank you for recognition of our work!
> > >
> > > Regarding curriculum learning conjecture, we are running ablation studies to verify this claim. Due to the congestion in our cluster, we are not sure if we can get the results before the end of the discussion period, but we'll try our best to give you an answer.

---

### Official Review · Reviewer_2kB4 · 2023-07-05

**Soundness:** 3 good
**Presentation:** 3 good
**Contribution:** 3 good
**Rating:** 6
**Confidence:** 3

**Summary:**

This paper presents a scalable path-based method for knowledge graph reasoning, which is inspired by the A* algorithm for shortest path problems.

**Strengths:**

1. The intriguing approach of applying the A$^*$ algorithm's principle to path reasoning in KG is proposed in this paper, along with the introduction of novel methods for crafting the priority function.

2. The paper achieves state-of-the-art results on the large-scale KG reasoning dataset, ogbl-wikikg2.

3. There's a substantial enhancement in efficiency, considering both time and memory usage, as opposed to the top-performing baseline, NBFNet.

**Weaknesses:**

The proposed method performs slightly worse than NBFnet as shown in Table 1, and no results of NBFnet are reported on tail prediction in Table 2.

**Questions:**

1. In the context of KG reasoning, a crucial question is, how many steps are typically required for a query? According to the vanilla path reasoning in Equation 1, the number of paths increases exponentially with respect to path length. However, if the path length is typically small, this might not pose a significant problem? Moreover, when dealing with a large-scale KG, the BF algorithm would need to visit $|\mathcal{V}|$ nodes and $|\mathcal{E}|$ edges for each step, which can be quite computationally intensive. Given these considerations, it leads to the question: If the path length is usually small, could vanilla path reasoning be a more efficient choice compared to BF?

2. Another question is, can we simply leverage the idea of beam search into vanilla path reasoning? For example, we keep top-K ranked paths for each step, which may also avoid the exponential growth of the number of paths.

**Limitations:**

Yes

---

> ### Author Rebuttal · Authors · 2023-08-08
>
> Thanks for your recognition and constructive comments. Here is our response to your concerns.
>
> **W1: The proposed method performs slightly worse than NBFNet as shown in Table 1, and no results of NBFNet are reported on tail prediction in Table 2.**
>
> A1: By design, A\*Net cannot be better than NBFNet in performance, and our goal is to show that the gap between A\*Net and NBFNet is very small (e.g. only 1% absolute difference in H@10 and less than 1% absolute difference in other metrics in Tab. 1).
>
> The reason is that A\*Net explores strictly less paths (e.g. 10% nodes and 10% edges on FB15k-237) than NBFNet in both training and inference. In a fair comparison, the performance of A\*Net should be roughly upper bounded by the performance of NBFNet. While it is possible to improve A\*Net by tweaking the neural parameterization and hyperparameters, it will result in an unfair comparison and blur the contribution of this paper.
>
> Here are the results including NBFNet on tail prediction. We can see that A\*Net achieves competitive performance compared to NBFNet on both datasets. We will update these results in the paper.
>
> |FB15k-237|MRR|H@1|H@3|H@10|
> |---|---|---|---|---|
> |MINERVA|0.293|0.217|0.329|0.456|
> |Multi-Hop|0.393|0.329|-|0.544|
> |CURL|0.306|0.224|0.341|0.470|
> |NBFNet|**0.509**|**0.411**|**0.562**|**0.697**|
> |A\*Net|**0.505**|**0.410**|**0.556**|0.687|
>
> |WN18RR|MRR|H@1|H@3|H@10|
> |---|---|---|---|---|
> |MINERVA|0.448|0.413|0.456|0.513|
> |Multi-Hop|0.472|0.437|-|0.542|
> |CURL|0.460|0.429|0.471|0.523|
> |NBFNet|**0.557**|**0.503**|**0.579**|**0.669**|
> |A\*Net|**0.557**|**0.504**|**0.580**|**0.666**|
>
> **Q1: In KG reasoning, how many steps are typically required for a query? If the path length is usually small, could vanilla path reasoning be a more efficient choice compared to Bellman-Ford?**
>
> A2: For path-based methods that use exhaustive search[1, 2, 3], they compute typically up to 3 steps due to poor scalability. For path-based methods that use the Bellman-Ford algorithm[4, 5], their ablation studies suggest that the performance keeps increasing with more steps until 6 steps.
>
> To give an intuition of how infeasible exhaustive search is, we compute the number of paths for different lengths, averaged over all positive triplets in each dataset. Here is the statistics. The number of paths grow exponentially w.r.t. the length of the paths. We note that exhaustive search is more efficient than the Bellman-Ford algorithm only when the number of paths is less than the number of nodes $|\mathcal{V}|$. This only holds for paths with length ≤ 1 on FB15k-237, length ≤ 4 on WN18RR and length ≤ 2 on ogbl-wikikg2 respectively. Therefore, vanilla path reasoning is usually not a good choice.
>
> ||\|V\||Length=1|Length=2|Length=3|Length=4|Length=5|Length=6|
> |---|---|---|---|---|---|---|---|
> |FB15k-237|14,541|367.0|31943|9.014e6|1.023e9|2.411e11|3.199e13|
> |WN18RR|40,943|19.38|138.9|4705.5|35504|1.624e6|1.214e7|
> |ogbl-wikikg2|2,500,604|135698|2.337e6|3.792e11|7.027e12|1.077e18|2.131e19|
>
> **Q2: Can we simply leverage the idea of beam search into vanilla path reasoning? For example, we keep top-K ranked paths for each step, which may also avoid the exponential growth of the number of paths.**
>
> A3: Beam search can be applied to path-finding methods[6, 7, 8] that use reinforcement learning to find paths. However, these methods assume a sparse set of answers (typically ≤100 answers) and can only be evaluated on tail prediction. Their performance is much worse than path-based methods that operate on a dense set of paths (e.g. NBFNet, A\*Net), as showed in Tab. 2.
>
> It is not trivial to apply beam search on path-based methods like NBFNet. First, beam search requires a score function to prune intermediate steps, which isn’t present in the original NBFNet. Second, even if we have such a score function, beam search is not differentiable w.r.t. the scores at the intermediate steps. This means we can’t learn the score function but can only use a handcrafted score function, which is likely to be suboptimal.
>
> [1] Lao and Cohen. Relational Retrieval Using a Combination of Path-Constrained Random Walks. ML 2010.
>
> [2] Neelakantan et al. Compositional Vector Space Models for Knowledge Base Completion. IJCNLP  2015.
>
> [3] Wang et al. Relational Message Passing for Knowledge Graph Completion. KDD 2021.
>
> [4] Zhu et al. Neural Bellman-Ford Networks: A General Graph Neural Network Framework for Link Prediction. NeurIPS 2021.
>
> [5] Zhang and Yao. Knowledge Graph Reasoning with Relational Digraph. WWW 2022.
>
> [6] Xiong et al. DeepPath: A Reinforcement Learning Method for Knowledge Graph Reasoning. ACL 2017.
>
> [7] Das et al. Go for a Walk and Arrive at the Answer: Reasoning Over Paths in Knowledge Bases using Reinforcement Learning. EACL 2017.
>
> [8] Lin et al. Multi-Hop Knowledge Graph Reasoning with Reward Shaping. EMNLP 2018.

---

> > ### Comment · Reviewer_2kB4 · 2023-08-17
> > **Thanks for your rebuttal**
> >
> > The authors addressed my concerns. Since my score is already positive, I'm maintaining my score.

---

### Official Review · Reviewer_7LW7 · 2023-07-06

**Soundness:** 3 good
**Presentation:** 3 good
**Contribution:** 2 fair
**Rating:** 4
**Confidence:** 4

**Summary:**

The main contribution of this paper is presenting a scalable path-based method A*Net, for link prediction on large-scale knowledge graphs. A*Net is inspired by the A* algorithm for solving shortest path problems, where it learns a priority function to select important nodes and edges at each iteration. This allows for the time and memory reducing for both training and inference. From an efficiency perspective, this could be considered as a path-pruning method to progressively reduce the subgraph based on the learned priority function. The empirical results also demonstrate efficiency improvement.

**Strengths:**

1. The efficiency problem caused by the explosively increasing entities in deeper propagation layers is indeed serious in the recent GNN-based inductive methods. And the proposed method makes sense and technically sound.

2. The experimental results are impressive. The paper demonstrates the practical applications of A*Net in various settings and datasets, with the efficiency improvement compared with several recent baselines. Furthermore, the paper sets a new state-of-the-art on the million-scale dataset ogbl-wikikg2 and converges faster than embedding methods.

3. The paper's organization is well-executed and the content is easily comprehensible.


**Weaknesses:**

1. The paper's comparison to the A* algorithm seems somewhat overstated. As a derivative work of NBFNet, this paper draws an analogy to another shortest path algorithm, A*. Contrary to the Bellman-Ford algorithm that resolves the shortest path problem from the source to all other points, the A* algorithm typically addresses the shortest path problem from the source to a specific target point. However, in the context of knowledge graph (KG) reasoning, the target point is unknown, rendering the core principle of A*, assessing the estimated remaining cost to the target point, unfeasible. In fact, the A* algorithm's priority rule, involving the distance to the target node, is not pertinent to the priority function in the proposed model. The A* algorithm appears to function primarily as a promotional point, rather than as a guiding principle.

2. Perhaps due to the overemphasis on the A* analogy, the paper's true technical contributions remain unclear. Comparing the core function of NBFNet in Eq. 3 and that of A*Net in Eq. 12, the only discernible difference lies in introducing the priority score, calculated based on the embeddings of the query and the current node. Stripping away the A* algorithm framework, it essentially seems to be a path-pruning technique reliant on an attention mechanism to select the top K nodes and top L edges in each layer for efficiency's sake.

3. The paper lacks insightful contributions regarding important paths beyond a weighted version of the NBENet method. The theoretical appendix focuses solely on integrating path selection into the NBFNet framework, premised on the assumption that a certain function can distinguish important nodes. However, how to ensure that important paths are chosen is not clear. In response to this, the authors propose weight sharing between the priority function and the predictor, asserting that the reasoning task can be seen as a weak supervision for the priority function. However, this appears counterintuitive, given that the priority score is dependent on a specific query. A high predictor score, indicating that the node x answers the query (u, r_1), should not contribute to the priority score of x for a different query (u, r_2).


**Questions:**

1. As addressed in Weaknesses 3, could you elaborate on how weight sharing aids in the selection of important paths?

2. I observe that two handcrafted priority functions, PPR and Degree, are employed in the ablation studies. Given that high connectivity doesn't necessarily denote the importance of paths, what about the effectiveness and efficiency of a random pruning strategy, particularly with respect to the obgl_wikikg2 dataset?

3. In the Visualization section, only the results of the proposed method are displayed without any comparison. Could you clarify what distinct paths the Neural function selects compared to the two handcrafted ones? Furthermore, does the Neural-based path selection align more closely with knowledge semantics?


**Limitations:**

Yes. The authors stated the limitation, future work and social impact.

---

> ### Author Rebuttal · Authors · 2023-08-08
>
> Thanks for your constructive comments. Here is our response to your concerns.
>
> **W1: In KG reasoning, the target point is unknown, rendering the core principle of A\* unfeasible. In fact, the A\* algorithm's priority rule, involving the distance to the target node, is not pertinent to the priority function in the proposed model.**
>
> A1: We agree that estimating the remaining distance is the core principle of the A\* algorithm. Our A\*Net follows this principle. We design Eqn. 10 in A\*Net to match Eqn. 4 in the A\* algorithm. $\mathbf{h}_q^{(t)}(u, x)$ corresponds to the current length $d(u, x)$, while $\mathbf{g}([\mathbf{h}_q^{(t)}(u,x), \mathbf{q}])$ corresponds to the remaining distance $g(x, v)$. A figure illustration is in the attached PDF file.
>
> Your misunderstanding might come from the fact that we don’t have $v$ in Eqn. 10. In fact, the learned representation $\mathbf{q}$ encodes the relative position from $u$ to $v$. e.g. If $q$ is the mother relation, the aggregation of paths between $u$ and $v$ in a positive sample should roughly match the representation of mother $\mathbf{q}$, i.e. $\mathbf{q} \approx \mathbf{h}^{(T)}_q(u, v)$ for any $(u,q,v)\in\mathcal{E}$. The reason why $\mathbf{q}$ can be independent of $u$ and $v$ is that the definition of a relation is independent of its triplet instances. The representation of the remaining cost is (due to TeX bugs, $g$ and $h$ should be $\mathbf{g}$ and $\mathbf{h}$ resp.)
>
> $g^{(t)}(x,v)\approx g([h_{q}^{(t)}(u,x),h_{q}^{(T)}(u,v)])\approx g([h_{q}^{(t)}(u,x),\mathbf{q}])$
>
> where the first approximation says that the remaining cost between $x$ and $v$ can be estimated by the current aggregation of paths $\mathbf{h}_{q}^{(t)}(u,x)$ between $u$ and $x$
>
> and the final path representations $\mathbf{h}_{q}^{(T)}(u,v)$ between $u$ and $v$.
>
> The second approximation replaces $\mathbf{h}_{q}^{(T)}(u,v)$ with $\mathbf{q}$. So $\mathbf{g}([\mathbf{h}_q^{(t)}(u,x), \mathbf{q}])$ matches the remaining distance $g(x, v)$ in the A\* algorithm.
>
> **W2: Due to the overemphasis on the A\* analogy, the paper's true technical contributions remain unclear.**
>
> A2: We clarified the analogy between A\*Net and the A\* algorithm in A1 and they align at each term in the priority function. Our major contribution is the first path-based method that scales to ogbl-wikikg2. Note that ogbl-wikikg2 has been previously dominant by embedding methods, and none of them are inductive or interpretable like A\*Net. Hence we think this contribution is very important to the community and may potentially change future research directions. The theoretical insights from the A\* algorithm are a minor contribution compared to our empirical achievements.
>
> **W3 & Q1: How to ensure that important paths are chosen is not clear. The weight sharing between the priority function and the predictor appears counterintuitive, given that the priority score is dependent on a specific query.**
>
> A3: Since there isn’t any annotation of the important paths, we can’t verify whether important paths are chosen or not. However, the observation that A\*Net matches the performance of NBFNet suggests that A\*Net captures most important paths. Note that the performance of A\*Net is upper bounded by NBFNet, as A\*Net visits much fewer paths.
>
> We agree that the priority score should be dependent on a specific query, and that’s what we designed A\*Net to be. Both the priority function and the predictor take a **query-dependent** representation $\mathbf{s}_{uq}^{(t)}(x)$ as input, but the parameters in both functions (i.e. $f(\cdot)$ in Eqn. 11) are **query independent**. This is consistent with previous works[1, 2, 3]. We share the **query-independent** parameters in these functions.
>
> For two queries $(u, q_1, ?)$ and $(u, q_2, ?)$, they will have different representations $\mathbf{s}_{uq_1}^{(t)}(x)$
>
> and $\mathbf{s}_{uq_2}^{(t)}(x)$, and hence different priority scores.
>
> **Q2: What about the effectiveness and efficiency of a random pruning strategy?**
>
> A4: We run experiments for the random pruning strategy on FB15k-237 and ogbl-wikikg2. We set the random pruning strategy to have the same node and edge ratios as A\*Net.
>
> A\*Net outperforms the random pruning strategy on both datasets, and is slightly better than the random pruning strategy in time and memory. We conjecture the reason is that A\*Net tends to revisit nodes more often than the random pruning strategy, resulting in a more cache-friendly behavior.
>
> |FB15k-237|MRR|H@1|H@3|H@10|#message|time|memory|
> |---|---|---|---|---|---|---|---|
> |Random|0.378|0.288|0.413|0.556|39,017|9.20min|16.9GiB|
> |A\*Net|**0.411**|**0.321**|**0.453**|**0.586**|38,610|8.07min|11.1GiB|
>
> |ogbl-wikikg2|TestMRR|ValidMRR|#Params|#message|time|memory|
> |---|---|---|---|---|---|---|
> |Random|0.5815|0.5827|**6.83M**|51,458|1.74hr|26.5GiB|
> |A\*Net|**0.6767**|**0.6851**|**6.83M**|52,371|1.30hr|24.1GiB|
>
> **Q3: Only the results of the proposed method are visualized without any comparison. What distinct paths the neural function selects compared to the two handcrafted ones? Does the neural-based path selection align more closely with knowledge semantics?**
>
> A5: Due to space limits, we put the visualization in the global response. We observe that the paths captured by A\*Net are concise. By contrast, NBFNet and PPR find longer and more redundant paths. The degree priority function results in the closest behavior as A\*Net, but the entities it visits are less relevant than those in A\*Net.
>
> It is hard to confirm that A\*Net aligns better with knowledge semantics, as we don’t have any ground truth for important paths. We don’t intend to claim the quality of the visualization of A\*Net, but just show its interpretability.
>
> [1] Zhu et al. A General Graph Neural Network Framework for Link Prediction. NeurIPS 2021.
>
> [2] Yang et al. Differentiable Learning of Logical Rules for Knowledge Base Reasoning. NIPS 2017.
>
> [3] Teru et al. Inductive Relation Prediction by Subgraph Reasoning. ICML 2020.

---

> > ### Comment · Reviewer_7LW7 · 2023-08-19
> >
> > Thanks so much for the detailed responses!
> > While the rebuttal does provide some clarification, I still have some concerns.
> >
> > 1. The authors explained the query vector $\mathbf{q}$ in the second term of Eq. 10 can represent the relative distance from $u$ to $v$. However, this interpretation seems to be more of an intuitive justification rather than an inherent aspect of the model design. Meanwhile, since $\mathbf{q}$ is also a factor in the computation of each $h^{(t)}_q(u,x)$, it remains unclear if $\mathbf{q}$ maintains its independence and intuitive significance throughout the training process.
> >
> > 2. The authors emphasized that the major contribution is the first path-based method that scales to ogbl-wikikg2. Nevertheless, I question the true significance of this experimental work for the inductive models calculating at subgraph level. Because the model would not calculate in the entire large-scale KG, excluding part of nodes/edges is an obvious strategy to improve efficiency and scalability. Compared with embedding-based methods, the natural drawback of inference complexity still exists.
> >
> > 3. I also have one additional concern on the model's performance. Table 1 illustrates that the proposed method performs slightly worse than NBFNet. I am skeptical of the statement that the performance of NBFNet represents a definitive upper bound. Notably, a recent study titled "AdaProp: Learning Adaptive Propagation for Graph Neural Network based Knowledge Graph Reasoning" has outperformed both RED-GNN and NBFNet by leveraging an edge sampling strategy.
> >
> > Given these considerations, I would like to keep my initial score.

---

> > > ### Author Response · Authors · 2023-08-20
> > > **Discussion (1/2)**
> > >
> > > Thank you for your feedback. Regarding your further concerns, here is our response. It would be nice if you can give us a basic feedback before the end of the discussion period.
> > >
> > > **C1: The query vector $\mathbf{q}$ in the second term of Eq. 10 can represent the relative distance from $u$ to $v$. However, this interpretation seems to be more of an intuitive justification rather than an inherent aspect of the model design. Meanwhile, since $\mathbf{q}$ is also a factor in the computation of each $h_q^{(t)}(u, v)$, it remains unclear if $\mathbf{q}$ maintains its independence and intuitive significance throughout the training process.**
> > >
> > > A1: We should correct that the query vector $\mathbf{q}$ represents the **relative position**, not the **relative distance**, from $u$ to $v$. The relative position contains not only the relative distance, but also the relative direction. For both A\*Net and the A\* algorithm, knowing the relative distance can’t fully recover the position of $v$ from the position of $u$, but knowing the relative position can. We agree that this is an intuitive justification, but it is also an inherent aspect for A\*Net to be inductive.
> > >
> > > We conjecture that you expected A\*Net to use the position of $v$ as the absolute goal, faithful to the A\* algorithm. First, as we disccused in Line 171-173, unlike the shortest path problem, we don’t know the answer entity $v$ beforehand in knowledge graph reasoning. So we can only reparameterize $v$ by $u$ and $q$ from the query. Second, if we use the absolute goal, no matter from an oracle or predicted by the representations $\mathbf{u}$ and $\mathbf{q}$, the model will fit to some absolute positional information, thereby can’t generalize to unseen entities in the inductive setting. Hence a relative goal $\mathbf{q}$ is better for preserving the inductive advantage of path-based methods.
> > >
> > > Yes, it is not that clear whether the representation $\mathbf{q}$ effectively captures the relative goal, when it is used as both the relative goal and the condition in Eqn. 2 & 3. We conduct ablation study to verify whether $\mathbf{q}$ contributes more as the relative goal or the condition. We consider two variants of A\*Net: 1) A variant without conditioning on $q$, where the indicator function (Eqn. 2) is replaced with $\mathbb{1}(u=v) = \overrightarrow{1}\text{ if }u=v\text{ else } \overrightarrow{0}$, and the edge representation (Eqn. 3) is replaced with $\mathbf{w}(x, r, v) = \mathbf{r}$. 2) A variant without the representation $\mathbf{q}$ in the neural priority function (Eqn. 10). Here are the results
> > >
> > > |FB15k-237|MRR|Hits@1|Hits@3|Hits@10|
> > > |---|---|---|---|---|
> > > |NBFNet (w/o conditioning)|0.400|0.306|0.439|0.585|
> > > |NBFNet|**0.415**|**0.321**|**0.454**|**0.599**|
> > > |A\*Net (w/o conditioning)|0.401|0.311|0.439|0.580|
> > > |A\*Net (w/o relative goal)|0.185|0.105|0.203|0.350|
> > > |A\*Net|**0.411**|**0.321**|**0.453**|0.586|
> > >
> > > We observe that conditioning on $q$ has a small gain on the performance (1.5% absolute difference in MRR) and is general to both NBFNet and A\*Net, suggesting that conditioning is not the key to the sucess of A\*Net. Note the conditioning design is a common practice to improve performance in path-based methods[1, 2, 3, 4, 5]. However, we observed a significant performance drop for A\*Net without $\mathbf{q}$ in the priority function (22.6% absolute difference in MRR). Hence we think $\mathbf{q}$ captures the relatively goal in the neural priority function and aligns with the intuition of the A\* algorithm.
> > >
> > > [1] Yang et al. Differentiable Learning of Logical Rules for Knowledge Base Reasoning. NIPS 2017.
> > >
> > > [2] Sadeghian et al. DRUM: End-To-End Differentiable Rule Mining On Knowledge Graphs. NeurIPS 2019.
> > >
> > > [3] Zhu et al. Neural Bellman-Ford Networks: A General Graph Neural Network Framework for Link Prediction. NeurIPS 2021.
> > >
> > > [4] Zhang and Yao. Knowledge Graph Reasoning with Relational Digraph. WWW 2022.
> > >
> > > [5] Zhang and Zhou et al. AdaProp: Learning Adaptive Propagation for Graph Neural Network based Knowledge Graph Reasoning. KDD 2023.

---

> > > ### Author Response · Authors · 2023-08-20
> > > **Discussion (2/2)**
> > >
> > > **C2: Because the model would not calculate in the entire large-scale KG, excluding part of nodes/edges is an obvious strategy to improve efficiency and scalability. Compared with embedding-based methods, the natural drawback of inference complexity still exists.**
> > >
> > > A2: It is always trivial to scale up models by random sampling and sacrificing the performance, but **it is not trivial if we can scale up models without performance drop**. This is also the main point of previous papers that studied sampling methods for GNNs on homogeneous graphs[6, 7]. As for A\*Net, we have shown in the answer to your Q2 that random pruning strategy is significantly worse than A\*Net on FB15k-237 (3.3% absolute difference in MRR) and ogbl-wikikg2 (9.5% absolute difference in MRR) under the same node and edge ratios. In other words, A\*Net is strictly better than random pruning in terms of efficiency and effectiveness.
> > >
> > > A\*Net has an inference complexity of $O(T(\alpha|\mathcal{V}|d^2+\alpha\beta|\mathcal{E}|d))$ for answering a single query $(u, q,?)$. By comparision, its full counterpart, NBFNet, has a complexity of $O(T(|\mathcal{V}|d^2+|\mathcal{E}|d))$. Embedding methods such as TransE or RotatE have a complexity of $O(|\mathcal{V}|d)$ for enumerating all entities to answer the query $(u, q,?)$. If we take the hyperparameters from ogbl-wikikg2, A\*Net is about $6\times(0.002\times2.5\text{M}\times32^2+0.002\times1\times16.1\text{M}\times32)=37\text{MFlops}$. Embedding methods are about $2.5\text{M}\times500=1.25\text{GFlops}$. Empirically, A\*Net is even faster than embedding methods due to its pruning ratios $\alpha$, $\beta$ and small dimension $d$.
> > >
> > > We would advocate to compare different knowledge graph reasoning methods through pareto frontiers, e.g. what is the best model at the complexity of $O(|\mathcal{V}|d)$, and what is the best model at the complexity of $O(|\mathcal{V}|^2d)$. A\*Net is a new pareto frontier here since it achieves competitive performance with NBFNet while uses signficantly less time. To our best knowledge, none of the embedding methods (e.g. TransE, RotatE) at the complexity of $O(|\mathcal{V}|d)$ is inductive, and none of the inductive methods (e.g. GraIL, NBFNet, A\*Net) can reach the complexity of $O(|\mathcal{V}|d)$.
> > >
> > > Also the inference complexity of embedding methods doesn’t suggest their actual time cost in applications. When the graph changes over time (e.g. Wikidata), embedding methods need to be frequently re-trained on the whole graph to accommodate any new entity, which is very costly. By contrast, inductive methods like A\*Net can directly perform inference over such new entities.
> > >
> > > **C3: I am skeptical of the statement that the performance of NBFNet represents a definitive upper bound. Notably, a recent study titled "AdaProp: Learning Adaptive Propagation for Graph Neural Network based Knowledge Graph Reasoning" has outperformed both RED-GNN and NBFNet by leveraging an edge sampling strategy.**
> > >
> > > A3: We feel it is a standard practice to assume that a sampling method is upper bounded by a full inference method in performance[6, 7] when they are compared as apples to apples. A\*Net exactly follows the neural parameterization and hyperparameters of NBFNet, so it is natural to think that A\*Net is upper bounded by NBFNet.
> > >
> > > AdaProp is compared to its full variant RED-GNN with different sets of hyperparameters, rather than a fair comparsion like ours. For example, AdaProp uses 8 layers and RED-GNN uses 5 layers, which explains the performance gain of AdaProp.
> > >
> > > [6] Chen et al. FastGCN: Fast Learning with Graph Convolutional Networks via Importance Sampling. ICLR 2018.
> > >
> > > [7] Chen et al. Stochastic Training of Graph Convolutional Networks with Variance Reduction. ICML 2018.

---

### Official Review · Reviewer_rgXX · 2023-07-12

**Soundness:** 2 fair
**Presentation:** 2 fair
**Contribution:** 2 fair
**Rating:** 5
**Confidence:** 3

**Summary:**

This paper proposes a scalable path-based knowledge graph reasoning approach. The idea is to extend only important paths from the exponentially growing set of all possible paths. A heuristic priority function is parametrized by a feed-forward network and is trained to predict the priority of nodes to expand. Experiments show that the proposed approach can significantly improve time and memory efficiency and also achieve good results.

**Strengths:**

- Scalability is an important issue for path-based reasoning approaches. The idea of selecting only important paths is interesting and sounds reasonable
- The proposed approach is effective and supported by extensive experiments. Time and memory efficiency has been significantly improved. Benchmark results are also good.

**Weaknesses:**

My concern is mainly about the design of the priority function Eq (10)

In Eq (10), the first part $h_q^{(t)}(u, x)$ is already conditioned on q, u, and x, so in principle the second part $g([h_q^{(t)}(u, x), q])$ doesn't provide any additional information. Therefore, the priority function is purely based on the current path from the start and contains no information about the goal. In other words, the prediction of the priority function would be the same even if the goal changes. This is different from the design of the A* algorithm and may lose theoretical guarantees.

It is not appropriate to present the approach in the manner of A* algorithm

**Questions:**

Please see Weaknesses

**Limitations:**

properly addressed

---

> ### Author Rebuttal · Authors · 2023-08-08
>
> Thanks for your comments. We notice that you raised only one concern for the rating of 3. Feel free to bring up other concerns for discussion. Here is our response to your concern.
>
> **W1: The first part $h^{(t)}_q(u, x)$ is already conditioned on q, u, and x, so in principle the second part $g([h^{(t)}_q(u, x), q])$ doesn't provide any additional information. The priority function is purely based on the current path from the start and contains no information about the goal. The prediction of the priority function would be the same even if the goal changes. This is different from the design of the A\* algorithm and may lose theoretical guarantees.**
>
> A1: It is a misunderstanding that the priority function contains no information about the goal. For a given query $(u,q,?)$, the goal is the set of answer nodes $\mathcal{V}_{ans} = \\{ v | (u,q,v) \in \mathcal{E} \\}$.
>
> Since the set $\mathcal{V}_{ans}$ is a function of $u$, $q$ and $\mathcal{E}$, we can represent the goal with source node $u$ and the query relation $q$. Note that we exclude the ground truth triplets $\mathcal{E}$ because they are unknown during inference. A more intuitive figure can be found in the PDF file attached to the global rebuttal response.
>
> The vector representation $\mathbf{s}_{uq}^{(t)}(x)$ for the priority function in A\*Net (Eqn. 10) is designed to match the priority function $s(x)$ in the A\* algorithm (Eqn. 4).
>
> For a given $u$, $x$ and $t$, $\mathbf{s}_{uq}^{(t)}(x)$ will be different if we change the goal $q$. We will make the correspondence between A\*Net and the A\* algorithm more explicit in the paper.
>
> Here we illustrate the correspondence between the terms in A\*Net and the A\* algorithm. The priority function in the A\* algorithm (Eqn. 4) is
>
> $$s(x) = d(u,x) \otimes g(x,v)$$
>
> where $d(u, x)$ is the length of the current shortest path from $u$ to $x$. $g(x, v)$ is a heuristic function that estimates the remaining length from $x$ to the target node $v$ (i.e. the goal). Typically, $g(x,v)$ is defined as the $L_1$ distance from $x$ to $v$.
>
> The vector representation $\mathbf{s}_{uq}^{(t)}(x)$ in A\*Net (Eqn. 10) is
>
> $$\mathbf{s}_{uq}^{(t)}(x) = \mathbf{h}_q^{(t)}(u,x) \otimes \mathbf{g}([\mathbf{h}_q^{(t)}(u,x), \mathbf{q}])$$
>
> where $\mathbf{h}_q^{(t)}(u, x)$ corresponds to $d(u, x)$ and represents the aggregation of paths from $u$ to $x$ within $t$ hops. The optional subscript $q$ means that the representation $\mathbf{h}_q^{(t)}(u, x)$ is conditioned on the goal $q$. In other words, $\mathbf{h}_q^{(t)}(u, x)$ mostly aggregates paths that are relevant to the goal $q$. One may also opt for an unconditioned version $\mathbf{h}^{(t)}(u, x)$ which aggregates all the paths uniformly. Here we follow NBFNet[1] and use this conditioned parameterization.
>
> We agree with you that the second part $\mathbf{g}([\mathbf{h}_q^{(t)}(u,x), \mathbf{q}])$ is conditioned on the same set of variables $u$, $q$ and $x$ as the first part. However, two architectures that model the same information don’t suggest they have the same inductive bias and generalization ability. For example, CNNs are better than MLPs on images because they have the inductive bias for translation equivariance, despite that they take the same information as input[2]. In graph machine learning, several papers[3, 4, 5] also suggest algorithmic alignment is the key to the success of GNNs on graphs.
>
> Here we show why $\mathbf{g}([\mathbf{h}_q^{(t)}(u,x), \mathbf{q}])$ is aligned with $g(x,v)$ in the A\* algorithm. The learned representation $\mathbf{q}$ captures the relative position from the source node $u$ to the target node $v$. For example, if $q$ is the mother relation, then the aggregation of paths between the source node $u$ and an answer node $v$ should roughly match the representation of mother $\mathbf{q}$, i.e. $\mathbf{q} \approx \mathbf{h}^{(T)}_q(u, v)$ for any $(u,q,v)\in\mathcal{E}$. The reason why $\mathbf{q}$ can be independent of $u$ and $v$ is that the definition of a relation is independent of its triplet instances. $\mathbf{h}_q^{(t)}(u,x)$ is the current aggregation of paths from $u$ to $x$ and represents the relative position from $u$ to $x$, conditioned on the query relation $q$. By transforming $\mathbf{h}_q^{(t)}(u,x)$ and $\mathbf{q}$ with a function $g(\cdot)$ (e.g. a vector substraction function), we can obtain an estimate of the relative position from $x$ to $v$, which corresponds to $g(x,v)$.
>
> [1] Zhu et al. Neural Bellman-Ford Networks: A General Graph Neural Network Framework for Link Prediction. NeurIPS 2021.
>
> [2] Bronstein et al. Geometric Deep Learning: Grids, Groups, Graphs, Geodesics, and Gauges. arXiv 2021.
>
> [3] Xu et al. What Can Neural Networks Reason About? ICLR 2020.
>
> [4] Xu et al. How Neural Networks Extrapolate: From Feedforward to Graph Neural Networks. ICLR 2021.
>
> [5] Dudzik and Velickovic. Graph Neural Networks are Dynamic Programmers. NeurIPS 2022.

---

> ### Comment · Reviewer_rgXX · 2023-08-21
> **Comment by Reviewer rgXX**
>
> I thank the authors for their detailed response
>
> I agree that $u$ and $q$ contain some information about the goal. Now the explanation is a bit clearer by regarding $q$ as some kind of "relative goal". But I still don't think the alignment between A\*Net and A\* algorithm is appropriate
>
> **In A\* algorithm, $d(u, x)$ and $g(x, v)$ model complementary information based on different inputs: the current shortest path from $u$ to $x$, and the estimation of the cost from $x$ to $v$, respectively. However, in A\*Net, $h_q(\cdot)$ and $g(\cdot)$ in Eq (10) are conditioned on the same variables $u$, $q$, $x$.** The authors' response argued that $\textbf{h}_q(\cdot)$ and $\textbf{g}(\cdot)$ can have different inductive biases, but this argument is too vague... It is still not clear enough what different information is modeled by $\textbf{h}_q(\cdot)$ and $\textbf{g}(\cdot)$, since $\textbf{h}_q(\cdot)$ also takes into account the "relative goal" $q$. Therefore I don't think there is a clear correspondence between A\*Net and A\* algorithm.
>
> I will raise my score to 4 to reflect the authors' further explanation

---

> > ### Author Response · Authors · 2023-08-21
> > **Discussion**
> >
> > Thank you for your feedback. We understand your concern that it looks like we fabricated the story to make A\*Net looks like the A\* algorithm, but they are actually well aligned. The basic logic behind the alignment is
> >
> > 1. The A* algorithm have two equivalent formulations, one based on the absolute goal $v$, and one based on the relative goal from $u$ to $v$, as illustrated in our attached PDF file.
> > 2. A\*Net follows the relative goal formulation of the A\* algorithm. There are two reasons that we must use this design: 1) We don’t have access to the absolute goal $v$ in knowledge graph reasoning, so we reparameterize with the source node $u$ and the relative goal $q$. 2) An absolute goal can’t transfer to unseen entities in the inductive setting, while a relative goal can.
> > 3. The reason that $\mathbf{h}_q(\cdot)$ and $\mathbf{g}(\cdot)$ are conditioned on the same variable is that we additionally condition the representation $\mathbf{h}_q(\cdot)$ on the query relation $q$ following the common practice in path-based methods[1, 2, 3, 4, 5]. This step is optional. In other words, we can think of that $\mathbf{h}(\cdot)$ is parameterized by $u$, $x$ and $\mathbf{g}(\cdot)$ is parameterized $u$, $x$, $q$.
> > 4. We verify this with ablation studies that either remove $q$ from the condition of $\mathbf{h}_q(\cdot)$ or from the priority function $\mathbf{g}(\cdot)$. We observe that conditioning on $q$ has a small gain on the performance (1.0% absolute difference in MRR) and is general to both NBFNet and A\*Net, suggesting that conditioning is not the key to the sucess of A\*Net. However, there is a significant performance drop for A\*Net without $\mathbf{q}$ in the priority function (22.6% absolute difference in MRR). This means that the relative goal $\mathbf{q}$ plays an important role in the priority function, which aligns with the intuition of the A\* algorithm.
> >
> >
> >     |FB15k-237|MRR|Hits@1|Hits@3|Hits@10|
> >     |---|---|---|---|---|
> >     |NBFNet (w/o conditioning)|0.400|0.306|0.439|0.585|
> >     |NBFNet|**0.415**|**0.321**|**0.454**|**0.599**|
> >     |A\*Net (w/o conditioning)|0.401|0.311|0.439|0.580|
> >     |A\*Net (w/o relative goal)|0.185|0.105|0.203|0.350|
> >     |A\*Net|**0.411**|**0.321**|**0.453**|0.586|
> >
> > We are aware that the alignment between A\*Net and the A\* algorithm is important for understanding the methodology of this paper. Hence we will clarify this point and add the above ablation studies in the revised version.
> >
> > [1] Yang et al. Differentiable Learning of Logical Rules for Knowledge Base Reasoning. NIPS 2017.
> >
> > [2] Sadeghian et al. DRUM: End-To-End Differentiable Rule Mining On Knowledge Graphs. NeurIPS 2019.
> >
> > [3] Zhu et al. Neural Bellman-Ford Networks: A General Graph Neural Network Framework for Link Prediction. NeurIPS 2021.
> >
> > [4] Zhang and Yao. Knowledge Graph Reasoning with Relational Digraph. WWW 2022.
> >
> > [5] Zhang and Zhou et al. AdaProp: Learning Adaptive Propagation for Graph Neural Network based Knowledge Graph Reasoning. KDD 2023.

---

### Author Rebuttal · Authors · 2023-08-09

# Summary of Responses

We would like to thank all reviewers for your time and patience on our submission. Here is a summary of reviewers’ points and our responses. **We attached a PDF file to illustrate the correspondance between A\*Net and the A\* algorithm.**

**Contributions**

- **Scalability is an important problem for path-based methods (all reviewers).**
- **A\*Net achieves impressive results in performance, time and memory, especially on a million-scale dataset ogbl-wikikg2 (all reviewers).**
- **The technical contribution of A\*Net is incremental (Reviewer 7LW7, ALvG):** The technical contribution involves mathematical derivation of A\*Net and the design of neural priority function inspired by the A\* algorithm. However, we emphasize the main contribution of this paper is to scale path-based methods to ogbl-wikikg2, which is **2 magnitudes larger** than datasets solved by previous path-based methods. This contribution is recognized by all reviewers. Considering that **ogbl-wikikg2 has been previously dominated by embedding methods**, this is an important breakthrough and may potentially change future research directions.

**Writing**

- **The neural priority function contains no information about the goal and is different from the A\* algorithm (Reviewer rgXX, 7LW7):** We design the representation for the neural priority function $\mathbf{s}_{uq}^{(t)}(x)$ in A\*Net (Eqn. 10) to match the priority function in the A\* algorithm (Eqn. 4). The current aggregation of paths $\mathbf{h}_q^{(t)}(u, x)$ corresponds to the current length $d(u, x)$ in the A\* algorithm, while the term $\mathbf{g}([\mathbf{h}_q^{(t)}(u,x), \mathbf{q}])$ corresponds to the remaining distance $g(x, v)$. While the A\* algorithm uses the target node $v$ as **the absolute goal**, A\*Net uses the learned vector $\mathbf{q}$ to represent **the relative goal** from the source node $u$ to the target node $v$. A more intuitive figure can be found in the attached PDF file.
- **The weight sharing between the priority function and the predictor is not clear. (Reviewer 7LW7, ALvG):** The priority function is $s_{uq}^{(t)}(x) = \sigma(f(\mathbf{s}_{uq}^{(t)}(x))$,

where $\mathbf{s}_{uq}^{(t)}(x)$ is the representation computed by Eqn. 10 and $f(\cdot)$ is a feed-forward network.

The predictor function is $p(v|u,q) = \sigma(f'(\mathbf{s}_{uq}^{(T)}(v)))$,

where $\mathbf{s}_{uq}^{(T)}(v)$ is the representation from the last layer and $f'(\cdot)$ is a feed-forward network. We share the parameters between $f(\cdot)$ and $f'(\cdot)$.

Note that $f(\cdot)$ is a query-independent function, but the node priority $s_{uq}^{(t)}(x)$ can be query dependent since the input representation $\mathbf{s}_{uq}^{(t)}(x)$ is dependent on the query relation $q$.
- We will carefully improve our writing to address these concerns in the revised version. Since these concerns are related to our technical contribution, please let us know if you have further questions.

**Experiments**

- **Comparison with a random pruning strategy (Reviewer 7LW7).**
- **Comparison on path visualization of A\*Net, NBFNet, and handcrafted priority functions (Reviewer 7LW7).**
- **Performance of NBFNet on tail prediction (Reviewer 2kB4).**
- **Performance of A\*Net when pruning is disabled (Reviewer ALvG).**
- We provide results to all the experiments required by the reviewers. All these experiments are consistent with the observations and claims in the paper. We will include them in the revised version to provide a better context to the readers.

**Questions**

- **If the path length is small, can vanilla path-based methods be more efficient than Bellman-Ford algorithm (Reviewer 2kB4):** We compute the number of paths for different lengths, averaged over all positive triplets in each dataset. We observe an exponential growth in the number of paths w.r.t. the length of the paths. Vanilla path-based methods are only efficient for length ≤ 1 on FB15k-237, length ≤ 4 on WN18RR and length ≤ 2 on ogbl-wikikg2 respectively. This is shorter than the optimal length of 6 reported by NBFNet and RED-GNN.

---

# Path Visualization of Different Methods

1. $(\text{Bandai}, \text{industry}, \text{?})$
- NBFNet
    - $\text{Bandai} \xleftarrow{\text{subsidiary}} \text{Bandai Namco Holdings} \xrightarrow{\text{webpage}} \text{official website}\xleftarrow{\text{webpage}}\text{Bandai Namco Entertainment}\xrightarrow{\text{industry}}\text{video game}$
    - $\text{Bandai}\xrightarrow{\text{webpage}}\text{official website}\xleftarrow{\text{webpage}} \text{Santa Clara}\xleftarrow{\text{location}} \text{Bandai Namco Entertainment}\xrightarrow{\text{industry}} \text{video game}$
- A\*Net (neural)
    - $\text{Bandai} \xleftarrow{\text{subsidiary}}\text{Bandai Namco Entertainment} \xrightarrow{\text{industry}}\text{video game}$
    - $\text{Bandai} \xrightarrow{\text{industry}}\text{media}\xleftarrow{\text{industry}}\text{Pony Canyon}\xrightarrow{\text{industry}}\text{video game}$
- A\*Net (PPR)
    - $\text{Bandai}\xrightarrow{\text{webpage}}\text{official website}\xleftarrow{\text{webpage}}\text{Bandai}\xrightarrow{\text{webpage}}\text{official website}\xleftarrow{\text{webpage}}\text{Bandai}\xleftarrow{\text{subsidiary}}\text{Bandai Namco Entertainment}\xrightarrow{\text{industry}}\text{video game}$
    - $\text{Bandai}\xrightarrow{\text{webpage}}\text{official website}\xleftarrow{\text{webpage}}\text{Bandai}\xleftarrow{\text{subsidiary}}\text{Bandai Namco Entertainment}\xrightarrow{\text{industry}}\text{video game}$
- A\*Net (degree)
    - $\text{Bandai}\xrightarrow{\text{webpage}}\text{official website}\xleftarrow{\text{webpage}}\text{Def Jam Recordings}\xrightarrow{\text{industry}}\text{video game}$
    - $\text{Bandai}\xleftarrow{\text{lead}}\text{CEO}\xrightarrow{\text{company}}\text{Microsoft}\xrightarrow{\text{industry}}\text{video game}$

---

> ### Author Response · Authors · 2023-08-20
>
> # Summary of Discussions
>
> We appreciate every reviewer who participated in the discussions. Here is a summary of updates in the discussions.
>
> **Previous concerns:**
>
> - **The technical contribution of A\*Net is incremental (Reviewer 7LW7, ALvG):**
>     - Reviewer ALvG doesn’t have further concerns on this point. Reviewer 7LW7 thinks sampling is an obvious strategy to improve scalability, and the time complexity of path-based methods is worse than embedding methods.
>     - We emphasize that it is not trivial to perform sampling without performance drop. In our experiments for Q2 of Reviewer 7LW7, A\*Net outperforms the random pruning strategy with similar time and memory by 9.5% absolute difference in MRR on ogbl-wikikg2. For the time complexity, A\*Net is empirically better than embedding methods on ogbl-wikikg2 due to the pruning ratios and small hidden dimensions. The complexity of embedding methods doesn’t fully reveal their actual time cost in applications, as they are not inductive and need to be re-trained when the graph changes.
> - **The neural priority function contains no information about the goal and is different from the A\* algorithm (Reviewer rgXX, 7LW7):**
>     - Both reviewers question about the necessity of the query vector $\mathbf{q}$ as the relative goal in the priority function, especially given that $\mathbf{q}$ is also used as the condition in the path representations.
>     - By design, A\*Net requires a relative goal, rather than an absolute goal in the A\* algorithm, in its priority function to be inductive. We conduct ablation studies for the representation $\mathbf{q}$ used as the condition and the relative goal on FB15k-237. Using $\mathbf{q}$ as a relative goal is important for A\*Net (22.6% absolute difference in MRR), while using $\mathbf{q}$ as the condition only slightly improves the performance (1.0% absolute different in MRR). We are aware that this is important for understanding the alignment between A*Net and the A* algorithm. We will clarify this with ablation studies in the revised version.
> - **The weight sharing between the priority function and the predictor is not clear (Reviewer 7LW7, ALvG):** Both reviewers don’t have further concerns on this point. We’ll include our clarification in the revised version.
> - **Comparison with a random pruning strategy (Reviewer 7LW7):** Reviewer 7LW7 doesn’t have further concerns on this point.
> - **Comparison on path visualization of A\*Net, NBFNet, and handcrafted priority functions (Reviewer 7LW7):** Reviewer 7LW7 doesn’t have further concerns on this point.
> - **Performance of NBFNet on tail prediction (Reviewer 2kB4):** Reviewer 2kB4 doesn’t have further concerns on this point.
> - **Performance of A\*Net when pruning is disabled (Reviewer ALvG):** Reviewer ALvG doesn’t have further concerns on this point.
>
> **New concerns:**
>
> - **A full inference method may not be an upper bound for a sampling method. AdaProp is better than RED-GNN (Reviewer 7LW7):** We feel it is a standard practice to assume a full inference method is the upper bound[1, 2]. AdaProp improves the performance of RED-GNN because it uses more layers, but here we seek a fair comparison between A\*Net and NBFNet.
>
> [1] Chen et al. FastGCN: Fast Learning with Graph Convolutional Networks via Importance Sampling. ICLR 2018.
>
> [2] Chen et al. Stochastic Training of Graph Convolutional Networks with Variance Reduction. ICML 2018.

---

### Decision · Program_Chairs · 2023-09-21

**Decision:**

Accept (poster)

**Comment:**

This paper presents A*Net, a scalable path-based method for knowledge graph reasoning. A*Net is inspired by the A* algorithm for shortest path problems and learns a priority function to select important nodes and edges at each iteration, which aims to reduce time and memory consumption for both training and inference. The reviewers and authors had a few rounds of discussion during the author response period, and some reviewers also followed up with the AC afterwards. The AC also read the paper, reviews and discussions. Most concerns were resolved. The reviewers and AC applauded the paper for studying the scalability issue of path-based reasoning approaches as well as the proposed approach being supported by extensive experiments. The remaining concerns are mainly about the analogy between A* and A*Net, where the comparison to A* seems to make the paper's technical contributions/novelty less clear.  In addition, reviewers are skeptical of the response that A*Net can be faster than traditional KGE models at inference time as well as Figure 1 (lack of the implementation details to fairly evaluate the training time). The authors are strongly encouraged to include the newly added experiments and clarify the aspects discussed in the response period in the revised version.